# AmpliconReconstructor integrates NGS and optical mapping to resolve the complex structures of focal amplifications

Jens Luebeck[1,2], Ceyda Coruh[3], Siavash R. Dehkordi[2], Joshua T. Lange[4,5], Kristen M. Turner[5], Viraj Deshpande [2], Dave A. Pai[6], Chao Zhang [1], Utkrisht Rajkumar[2], Julie A. Law [3], Paul S. Mischel [5,7,8] & Vineet Bafna[2 ✉]

Oncogene amplification, a major driver of cancer pathogenicity, is often mediated through focal amplification of genomic segments. Recent results implicate extrachromosomal DNA (ecDNA) as the primary driver of focal copy number amplification (fCNA) - enabling gene amplification, rapid tumor evolution, and the rewiring of regulatory circuitry. Resolving an fCNA's structure is a first step in deciphering the mechanisms of its genesis and the fCNA's subsequent biological consequences. We introduce a computational method, AmpliconReconstructor (AR), for integrating optical mapping (OM) of long DNA fragments (>150 kb) with next-generation sequencing (NGS) to resolve fCNAs at single-nucleotide resolution. AR uses an NGS-derived breakpoint graph alongside OM scaffolds to produce high-fidelity reconstructions. After validating its performance through multiple simulation strategies, AR reconstructed fCNAs in seven cancer cell lines to reveal the complex architecture of ecDNA, a breakage-fusion-bridge and other complex rearrangements. By reconstructing the rearrangement signatures associated with an fCNA's generative mechanism, AR enables a more thorough understanding of the origins of fCNAs.

[1] Bioinformatics and Systems Biology Graduate Program, University of California at San Diego, La Jolla, CA 92093, USA. [2] Department of Computer Science and Engineering, University of California at San Diego, La Jolla, CA 92093, USA. [3] Plant Molecular and Cellular Biology Laboratory, Salk Institute for Biological Studies, La Jolla, CA 92037, USA. [4] Biomedical Sciences Graduate Program, University of California at San Diego, La Jolla, CA 92093, USA. [5] Ludwig Institute for Cancer Research, University of California at San Diego, La Jolla, CA 92093, USA. [6] Bionano Genomics, Inc., San Diego, CA 92121, USA. [7] Moores Cancer Center, University of California at San Diego, La Jolla, CA 92093, USA. [8] Department of Pathology, University of California at San Diego, La Jolla, CA 92093, USA. ✉email: vbafna@cs.ucsd.edu

Oncogene amplification is a major driver of cancer pathogenicity[1–5]. Genomic signatures of oncogene amplification include somatic focal copy number amplifications (fCNAs) of relatively short (typically <10 Mbp) genomic regions[5,6]. Multiple mechanisms cause fCNAs including, but not limited to, extrachromosomal DNA (ecDNA) formation[5,7,8], chromothripsis[9], tandem duplications[10,11], and breakage–fusion–bridge (BFB) cycles[12–14]. EcDNA, in particular, enables tumors to achieve far higher oncogene genomic copy numbers and maintain far greater levels of intratumor genetic heterogeneity than previously anticipated, due to their non-chromosomal mechanism of inheritance—enabling tumors to evolve rapidly[5,15,16]. In addition, the very high DNA template level generated by ecDNA-based amplification, coupled to its highly accessible chromatin architecture, permits massive oncogene transcription[17–19].

While ecDNA elements are a common form of fCNA[5], other mechanisms can also result in amplification with different functional consequences[6]. Accurate identification and reconstruction of the fCNA structure not only describes the rearranged genomic landscape, but also represents a first step in identifying the generative mechanism—to ultimately gain understanding about an fCNA's biological consequence. Reconstruction of fCNA architecture involves determining the order and orientation of the genomic segments that constitute the amplicon. There are many methods to detect single genomic breakpoints from sequencing data, using a variety of different sequencing technologies[20–24]. However, fewer methods are available to handle the more difficult problem of ordering and orienting multiple genomic segments joined by breakpoints into high-confidence copy number-aware scaffolds, which are subsequently joined to enable complete reconstructions of complex rearrangements[6,25]. This problem represents the key algorithmic challenge addressed by our work.

A previous method for characterizing the identity of focally amplified genomic regions, AmpliconArchitect (AA), generates an accurate breakpoint graph from next-generation sequencing (NGS) data[6]. The graph encodes the genomic segments involved in fCNAs, their copy numbers, and breakpoint edges connecting the segments. Unambiguous reconstruction of fCNA architecture requires extracting paths and cycles from the breakpoint graph, to reveal the true structure of the underlying rearranged genome. However, in practice, path/cycle extraction is often confounded by duplications of large genomic regions inside an amplicon (Supplementary Fig. 1a), imperfections in the graph arising from errors in estimation of segment copy numbers, erroneous and/or missing breakpoints.

We hypothesized that an approach combining the strengths of NGS with long-range genome mapping data would enable larger and more unambiguous reconstructions of fCNA architectures. We utilized optical mapping (OM) data, which provides single-molecule information about the approximate locations of fluorescently-labeled sequence motifs on long fragments of DNA[26]. Importantly, optical mapping has orthogonal sources of error to DNA sequencing[27,28]. Primary sources of error to consider include missing OM labels, uncertainty about the exact location of the label on the imaged molecule, and possible molecular chimerism. The median molecule (map) length used in assembly across all samples present in this study is 244 kbp (molecule N50 340 kbp), while the median segment length in breakpoint graphs in this study is 100 kbp, highlighting that OM data can span multiple junctions in breakpoint graphs derived from focal amplifications (Supplementary Data 1). The integrated NGS data and OM data provide an orthogonal pairing of short- and long-range information about genomic structural variation.

We present a computational method for reconstructing large complex fCNAs, AmpliconReconstructor (AR). AR takes a breakpoint graph and long-range OM data as inputs. We utilize Bionano (Bionano Genomics, Inc., San Diego, CA) whole-genome imaging to generate single-molecule optical maps, which are de novo assembled into OM contigs (contig N50 72.8 Mbp). AR produces an ordering and orientation of graph segments, with fine-structure information from the breakpoint graph embedded into the large-scale reconstructions. As output, AR reports large-scale reconstructions of fCNA amplicons. We demonstrate the large-scale and fine-scale accuracy of AR using simulated OM data derived from seven cancer cell lines[6,21] (CAKI-2, GBM39, NCI-H460, HCC827, HK301, K562, and T47D). Finally, we validate the fCNA reconstructions using cytogenetics.

## Results

**Overview of AR**. We formulated the problem of fCNA reconstruction in multiple parts. First, alignment of genomic segments with optical map contigs. Second, the reconstruction of a genomic scaffold using OM data as a backbone. Third, the identification of the maximal simple paths in a graph where each node is an OM scaffold, for which the path is not a subsequence of another maximal simple path. AR separates these computational tasks into four primary modules (Fig. 1a, b). To address the first problem, we designed an OM alignment module, SegAligner, for aligning reference segments to assembled OM contigs generated by either the Bionano Irys or Bionano Saphyr instruments (Supplementary Fig. 1b–d, Methods—"Optical map contig alignments with SegAligner"). SegAligner is critical as it can score placements of short genomic segments onto an OM contig, which was not possible with other aligners. To address the second problem, we introduce two modules. First, a scaffolding module, which takes a collection of breakpoint graph segments aligned to OM contigs as input and creates scaffolds represented by directed acyclic graphs (DAGs) (Fig. 1c–e, Methods—"Reconstructing amplicon paths with AR"). The second module for scaffolding with AR involves a novel scaffold-path imputation technique (Fig. 1f–h, Methods—"Imputing paths in the scaffold with AR") to connect breakpoint graph segments that may individually be too small to be informatively labeled and aligned with optical mapping (Fig. 1f). We address the final problem with a path-finding module, which links scaffolds and searches for paths in a copy number (CN)-aware manner, to identify possible reconstructions of the amplicon. AR outputs a collection of sequence resolved paths supported by the linked scaffolds. We implemented a visualization utility, CycleViz, to show the integrated OM- and NGS-derived breakpoint graph data (Supplementary Fig. 2). AR is implemented in Python, and SegAligner is implemented in C++. Both tools are available publicly at https://github.com/jluebeck/AmpliconReconstructor.

**AR accurately reconstructs simulated amplicons**. We utilized multiple simulation strategies to measure the performance of AR (Supplementary Fig. 3). We used 85 non-trivial breakpoint graph paths reported by AmpliconArchitect from 25 cancer cell lines[6] as a ground-truth set of amplicon structures, and a separate simulation of 20 de novo simulated circular ecDNA structures. We first present the results of the 85 breakpoint graph paths. These paths included cyclic (37 paths) and non-cyclic paths (48 paths) with lengths varying from 260 kbp to 2.8 Mbp (median 1.1 Mbp) and the number of graph segments varying from 3 to 47 (mean 17.5 segments; Supplementary Data 2). These paths were used as a reference from which we simulated OM molecules (Methods—"Simulation of amplicons to measure AR performance"). Simulated molecules were assembled into contigs using the Bionano Assembler[29,30].

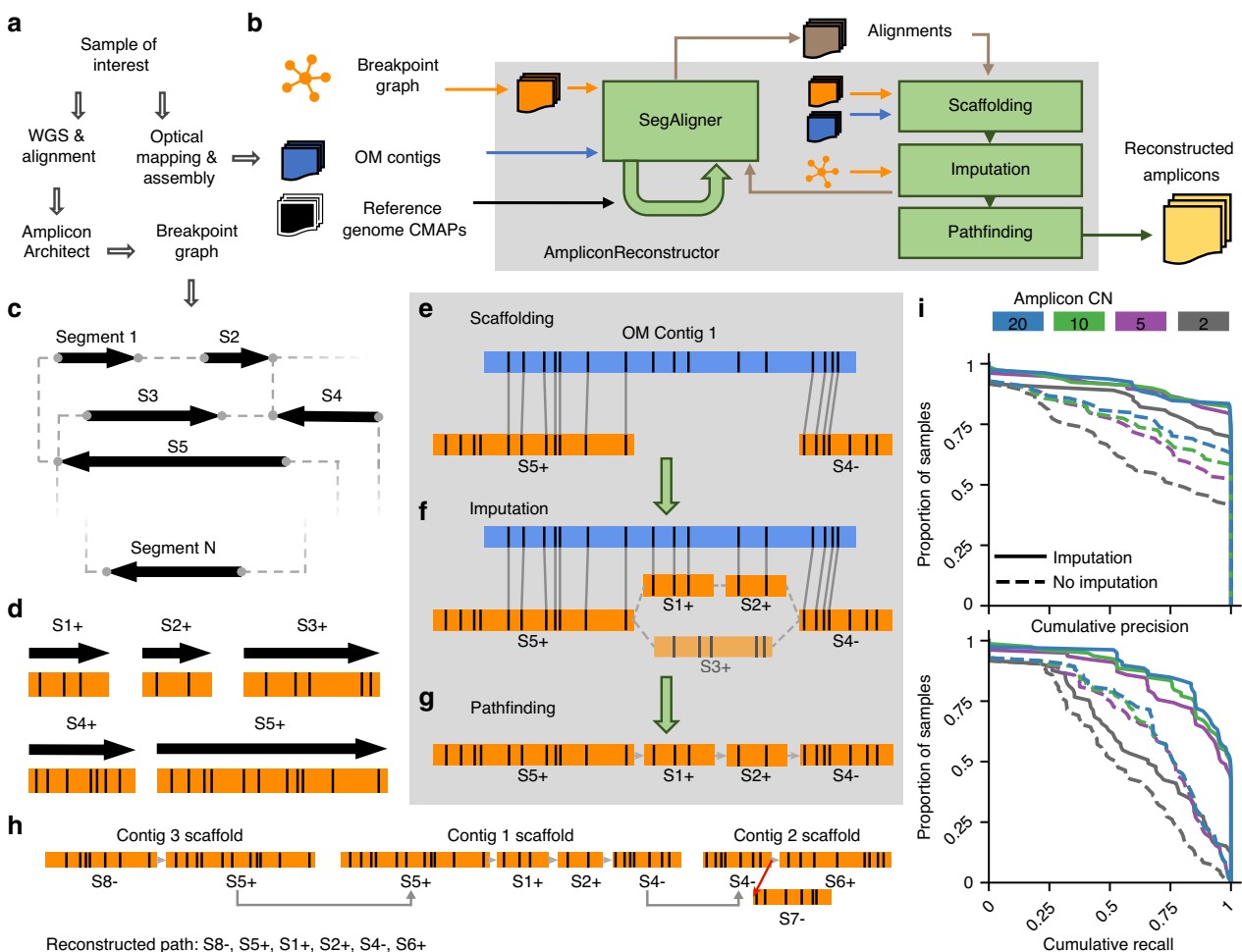

**Fig. 1 AmpliconReconstructor (AR) overview. a** Workflow to produce the necessary inputs for AR. AR accepts OM data in the consensus map (CMAP) format. **b** High-level overview of the AR method, where the inputs and outputs are shown outside the gray box representing the AR wrapper. The green loop-back arrow on the SegAligner module represents the identification of reference segments not encoded in the breakpoint graph. **c** A breakpoint graph with $N$ segments. **d** In silico digestion of breakpoint graph segments (orientation given by $+/-$) from (**c**) to produce graph OM segments. **e** Alignment of graph OM segments to OM contigs produces a scaffold of segment-contig alignments. **f** AR uses the structure of the breakpoint graph to identify paths between scaffold alignment endpoints which are also paths in the breakpoint graph. AR generates composite optical maps from combined path segments to score each candidate path against the gap in the scaffold. **g** AR identifies a candidate path with maximum score out of the possible imputed paths between two alignments. **h** AR links individual scaffolds sharing overlap between graph segments. The resulting graph has two types of edges, allowed (gray) and forbidden (red). **i** Cumulative precision and recall curves based on simulated OM data for AR using SegAligner, calculated with the Length (bp) LCS metric. Line color indicates the copy number (CN) of the simulated amplicon. Source data are provided as a Source data file.

For each of the 85 simulated amplicons, we ran AR on the corresponding breakpoint graph and the de novo assembled contigs, and examined four different variables that could affect the performance of AR. First, we tested AR performance using SegAligner for OM alignment, versus AR using other OM alignment tools to replace SegAligner. Second, we evaluated the performance of AR across a range of amplicon copy numbers. Third, we measured performance with false edges present in the breakpoint graph. Finally, we generated and tested mixtures of three similar amplicons from the same samples, simulated with different amplicon copy numbers, to measure the effects of potential amplicon heterogeneity on AR performance.

We measured the accuracy of AR by computing precision and recall across the four simulation conditions. As precision and recall could be quantified in multiple ways when comparing ground-truth and reconstructed simulation paths, leading to different understandings of performance, we described three ways of measuring the similarity of the paths (Length (bp), Nseg, Breakpoint; Methods—"Measuring AR simulation

performance"), based on the longest common substring (LCS) between ground-truth and reconstructed path sequences. We report the Length (bp) measurement in the analysis described here, while results with other measurements are presented in Supplementary Data 2 and Supplementary Fig. 4.

AR using SegAligner achieved a mean F1 score (harmonic mean of the precision and recall) of 0.88 for the highest copy number level (CN 20) and 0.68 for the lowest copy number level (CN 2) (Fig. 1i, Supplementary Fig. 4, Supplementary Data 2). In contrast, when OMBlast[31] or Bionano RefAligner[29,32] were used in place of SegAligner, we noticed a decrease in both precision and recall. For RefAligner and OMBlast, respectively, we report mean F1 scores of 0.52, 0.43 for CN 20, and 0.42, 0.41 for CN 2. When imputation was omitted from AR, the mean F1 score for CN 20 decreased from 0.88 to 0.70. We observed similarly consistent trends using other methods of measuring precision and recall—Nseg and Breakpoint (Supplementary Fig. 4). Large duplications inside a rearranged amplicon represent a challenging case to reconstruct. We identified 60 duplications of one or more

graph segments (mean length 281 kbp) in the simulated amplicons, and we report that AR resolved 75% (45) of these duplications. We saw some cases of 'assembly failure,' where no paths differing from the reference genome involving the amplicon segments were assembled. Figure 1i shows cumulative precision and recall values for AR using SegAligner (with and without imputation), and with assembly failures filtered. We additionally reported simulation F1 scores with and without filtering for assembly failure (Supplementary Data 2).

To understand the reasons for loss of performance on a small number of simulation cases, we examined the results from the CN 20 simulation where individual reconstructions showed either precision or recall <0.6. We manually examined the results from the 85 total cases and found that of 13 amplicons with precision below this threshold, nine cases showed signs of assembly failure, while three had incorrect reconstructions likely on account of graph complexity. The remaining case showed an issue with incorrect scaffold linking. Of 14 amplicons having recall below the threshold, nine cases showed signs of assembly failure, while five had highly segmented breakpoint graphs making it difficult for AR to identify anchoring alignments around the breakpoints, leading to an incomplete reconstruction.

False edges in the breakpoint graph increase the possible number of path imputations that AR considers, potentially leading to erroneous scaffolds. On simulated CN 20 amplicons, we added additional false edges between existing graph segments. We tested three scenarios with the proportion of additional false edges ranging from 0, 50, and 100% of the number of true graph edges. The three scenarios resulted in nearly identical mean F1 scores of 0.881, 0.880, 0.881 across the 85 amplicon simulations (Supplementary Data 2, Supplementary Fig. 5a), highlighting the robustness of the path imputation method.

To understand how AR performed when faced with amplicon heterogeneity, we designed a simulation study involving 123 combinations of breakpoint graph paths where each combination was derived from paths found in a single sample, generated at varying copy number mixtures. We simulated amplicons from heterogeneous mixtures with (1) a single dominant amplicon (CNs 20-2-2); (2) a linear mixture of CNs (CNs 20-15-10); (3) equally abundant amplicons (CNs 20-20-20). We report mean F1 scores of 0.92, 0.89, and 0.91, respectively, for the three cases (Supplementary Data 2). To explain the increase in performance of the mixture simulations as compared to the single amplicon simulations, we hypothesize that the greater total number of molecules improved the assembly process. Regardless, the high similarity between the precision and recall in each mixture case (Supplementary Fig. 5b) indicates AR can reconstruct an accurate amplicon path even in the context of heterogeneity.

Last, we designed a simulation strategy not reliant on prior AA-generated paths. Instead, we generated 20 de novo simulated rearranged circular amplicons (median size 2.0 Mbp, mean segments 9.3) and replaced the hg19 reference used to generate background molecules with a simulated tumor genome generated with SCNVSim[33]. AR's performance on these cases achieved a mean F1 score of 0.860 (0.731 when assembly failures included, Supplementary Data 2). The distributions of F1 scores for the 20 de novo cases and the 85 AA-derived cases were not statistically different between the 85 AA-derived amplicons and the 20 de novo simulated amplicons (two-tailed Mann–Whitney U-test, $p$-value = 0.1996, test statistic = 631.0). Based on these results, we found AR to be robust, and to outperform other methods for resolving fCNA architecture.

**AR reconstructs ecDNA in multiple forms**. Three cell lines in our study were previously reported to contain ecDNA[5]—GBM39,

NCI-H460, and HK301. We previously analyzed[17] glioblastoma multiforme (GBM) cell line GBM39 using a preliminary version of AR with Bionano RefAligner[29,32] and manual merging of graph segments. Re-analysis reproduced an unambiguous 1.26 Mbp *EGFRvIII*-containing circular ecDNA identical to the previously published structure[17] (Supplementary Fig. 6). The entire amplicon was captured by a single non-circular OM contig, with circularity confirmed by an overlapping breakpoint graph segment aligned to both ends of the contig.

Prior studies of ecDNA have documented their integration into chromosomes over time, linearizing and appearing as homogeneously staining regions (HSRs), often in non-native locations[5,7,15]. In a previous study[5], The GBM cell line HK301 had been cytogenetically determined to have circular ecDNA; however, we observed from FISH (fluorescence in situ hybridization) data that the sample's ecDNA had become HSR-like at the time of this study (Fig. 2a). AA reported a breakpoint graph supporting amplification of both *EGFRvIII* and *EGFR* wild-type (Fig. 2c), however, an unambiguous reconstruction from the graph alone was not possible. The AR reconstruction of the HK301 fCNA indicated a complex cyclic structure supported by three contigs (Fig. 2d), which explained 98.1% of the amplified genomic regions. The graph segments came predominantly from chr7, but also included two small regions (2890 bp, 4591 bp) from chr6 (Fig. 2c, d). We noted a ~20 kbp deletion inside *EGFR*, showing a lower CN than the surrounding region, but which was still amplified over the baseline regions of chr7. This indicates heterogeneity of *EGFR* wild-type/vIII mutation status. Despite the heterogenous status of this allele, AR reconstructed the *EGFRvIII* version—which is the dominant form of the amplicon (Fig. 2d).

The lung cancer cell line NCI-H460 has previously been documented to bear *MYC* amplification[34], and our cytogenetic analysis showed evidence for both its HSR-like and ecDNA amplification (Fig. 2e, f). Despite the heterogeneous nature of the amplicon's integration status, AA generated a breakpoint graph for a contiguous 2.15 Mbp region of chr8 (Fig. 2g). AR reconstructed a single 4.10 Mbp structure supported by five OM contigs (Fig. 2h). This structure contained all amplified segments from the breakpoint graph and explained the relative ratios of breakpoint graph segment copy numbers. For example, segment chr8:129,404,278–129,591,422 appeared four times, chr8:128,690,200–129,404,277 (carrying *MYC* & *PVT1*) appeared twice, chr8:129,591,423–129,911,811 appeared twice, and chr8:129,911,812–130,640,594 appeared once, making the ratios consistent with the estimated graph segment copy numbers (46, 25, 25, 12, respectively; Fig. 2g). The status of the long non-coding RNA *PVT1* (a known regulator of *MYC*)[35] on this amplicon is heterogeneous, as one copy of *PVT1* does not contain breakpoints, while the other shows a disrupted copy of *PVT1*. AR also identified a self-inversion at the end of the amplicon (black arrows in Fig. 2h), suggestive of an alternating forward-backward orientation (segmental tandem aggregation with inversion) of the amplicon in the agglomerated ecDNA.

We previously documented a circular amplicon containing an integrated human papillomavirus-16 (HPV16) genome[6], and we hypothesized that AR could help resolve the location of viral insertion in the host genome. We simulated a 1 Mbp circular amplicon with the 7.9 kbp HPV16 genome randomly inserted. AR was able to reconstruct the circular ecDNA structure and identified the integration point of human papillomavirus-16 (Supplementary Fig. 7) despite the viral genome having no OM labeling sites, suggesting that AR would serve as useful method for validating the existence of genomic oncovirus integrations suggested by NGS data.

In summary, AR reconstructed paths that were consistent with the expected ratios between amplified segment copy numbers and

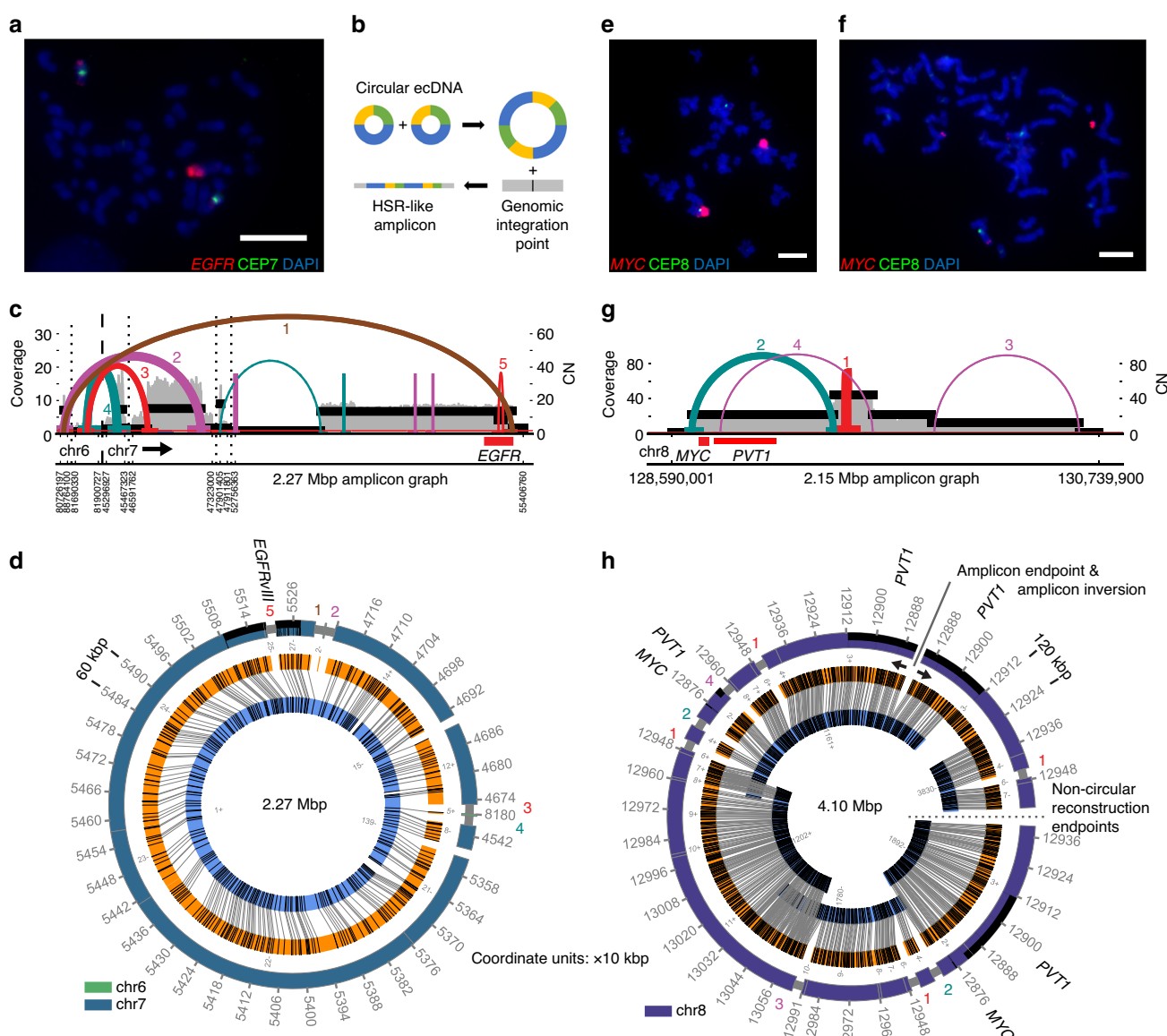

**Fig. 2 Reconstruction of extrachromosomal DNA (ecDNA). a** FISH with DAPI (4′,6-diamidino-2-phenylindole)-stained metaphase chromosomes in HK301 showing an HSR-like amplicon containing *EGFR*. Scale bar indicates 10 μm. **b** Theoretical model for the integration of circular extrachromosomal DNA into HSR-like amplicons, preserving the structure of breakpoint graph. **c** AA-generated breakpoint graph for HK301 containing *EGFR* and segments from chr6. The coloring of the graph edges represents the orientation of the junction between the two segments. Edge thickness indicates AA-estimated breakpoint copy number. Vertical dashed lines separate segments from different chromosomes while dotted lines distinct genomic regions from the same chromosome. Numbering of breakpoint edges corresponds with AR reconstruction breakpoint numbering. **d** Cyclic AR reconstruction of HK301 amplicon containing *EGFRvIII*. Breakpoint graph edges supported by the AA graph are numbered in a manner corresponding to the numbering in panel (**c**). **e** FISH with DAPI-stained metaphase chromosomes in NCI-H460 shows HSR-like *MYC* amplicon. Scale bar indicates 7.3 μm. **f** FISH with DAPI-stained metaphase chromosomes in NCI-H460 showing an extrachromosomal *MYC* amplicon. Scale bar indicates 7.3 μm. **g** AA-generated breakpoint graph for NCI-H460 containing *MYC* and *PVT1*. **h** AR reconstruction of the NCI-H460 amplicon. Indicated in this figure is an amplicon inversion point (top right) where the reconstruction explaining the full amplicon ends, and then the structure begins to repeat in the opposite direction (solid line & opposing black arrows). Also indicated is an endpoint for the non-circular reconstruction (center right) where the AR reconstruction and full amplicon structure both stop (dotted line).

graph structures in GBM39, HK301, and NCI-H460, explaining 99.9%, 98.1%, and 100% of the amplified genomic content in the breakpoint graphs for each cell line, respectively. Furthermore, the AR reconstructions of ecDNA in HSR-like form lend additional evidence to the agglomerative model of ecDNA integration (Fig. 2b)[8,36,37].

**AR reconstructs a rearranged Philadelphia chromosome in K562.** The classical model of the *BCR-ABL1* fusion involves a reciprocal translocation of the q arms of chromosomes 9 and 22

(Philadelphia chromosome)[38]. However, this mechanism alone does not explain the copy number amplification of *BCR-ABL1* fusion commonly observed in chronic myeloid leukemia (CML), highlighting a need for methods to better understand the genesis of the *BCR-ABL1* amplification[39,40]. To reconstruct the fine structure of a Philadelphia chromosome, we used the CML cell line K562 where a *BCR-ABL1* fusion had previously been reported[41].

The AA-reconstructed breakpoint graph for the *BCR-ABL1* fCNA in K562 contains 8.5 Mbp of amplified genomic segments (Fig. 3a). The graph shows signatures of complex rearrangements

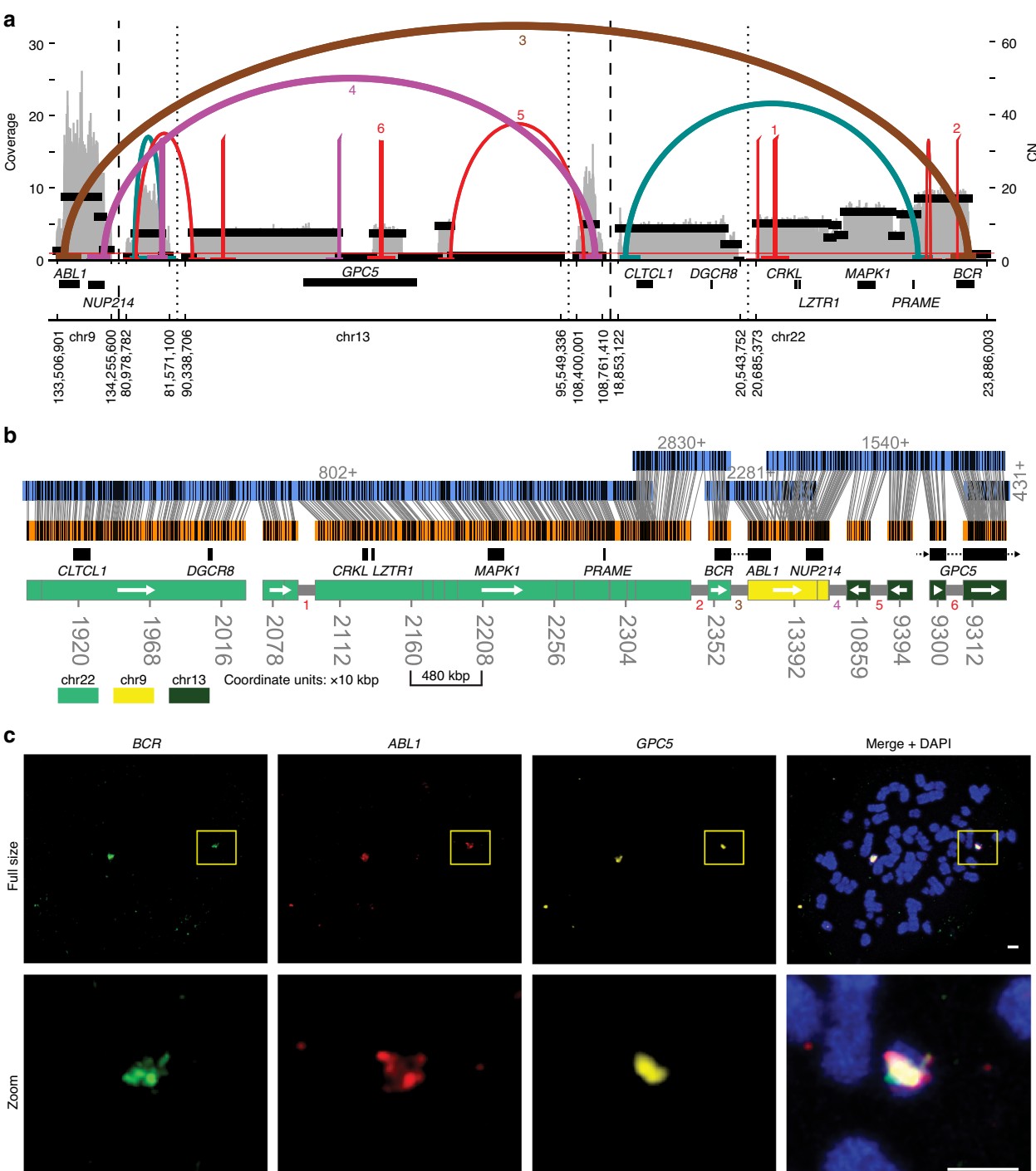

**Fig. 3 Reconstruction of a complex Philadelphia chromosome. a** AA-generated breakpoint graph for K562. Estimated copy number (CN), coverage, discordant reads forming breakpoint graph edges, and a subset of the genes in these regions are shown. **b** AR reconstruction of an 8.5 Mbp focal amplification which was supported by both Irys and Saphyr reconstructions. The tracks from top to bottom are: OM contigs (with contig ID and direction indicated above), graph segments (alignments shown with vertical gray lines), gene subset, and color-coded reference genome bar with genomic coordinates (scaled as 10 kbp units). Gray half-height bars between individual segments on the reference genome bar indicate support from edges in the AA breakpoint graph. White arrows inside the chromosome color bar indicate direction of genomic segment(s). Colored numbers correspond to numbered breakpoint graph edges in panel (**a**). **c** Multi-FISH using probes against *BCR*, *ABL1*, and *GPC5* with DAPI-stained metaphase chromosomes. Scale bars indicate 2 µm in both "Full size" and "Zoom" rows.

alongside the *BCR-ABL1* fusion, which AA predicted to have a copy number of 17 (Fig. 3a). We generated both Bionano Irys and Bionano Saphyr OM data for K562 cells and observed consistent results in the independent reconstructions of amplicons from both sources (Supplementary Fig. 8a, b). Using the breakpoint

graph and OM contigs, AR reconstructed a complex linear structure that chained together 1.7 Mbp from chr22 (containing *BCR*), 548 kbp of chr9 (containing *ABL1*), and multiple regions from chr13 (732 kbp; including a disrupted copy of *GPC5*) (Fig. 3b). In Fig. 3b, we show one possible scaffolding of the given

regions, whose structure was reproduced in both Saphyr and Irys datasets. AR also reported junctions between segments in the breakpoint graph where NGS-derived breakpoint edges were not reported, as indicated by the missing half-height gray bars between adjacent genomic segments in the genome tracks of Fig. 3b. While AR explains many of the amplified segments in this amplicon, we note that there is additional copy number variation in this amplicon it does not explain. For instance, the *BCR* and ABL-containing segments have an elevated CN over the segments on chr13.

We performed FISH experiments using combinations of probes for *BCR*, *ABL1*, *GPC5*, and chr22 centromere probe CEP22. The FISH images confirmed the co-localization of the *BCR-ABL1* fusion and *GPC5* on a common HSR-like structure (Fig. 3c) and validated the status of the *BCR-ABL1* fusion as being located on chr22 (Supplementary Fig. 9).

In addition to the reconstruction reported in Fig. 3b, AR identified other scaffolds, indicating that the genomic structure surrounding the *BCR-ABL1* translocation may be varied across the multiple copies (Supplementary Figs. 8c, d and 10a-f). In particular, the genomic segment bearing *CLTCL1* appears in both forward and reverse directions (Supplementary Fig. 10b, c). Other amplified regions of chr13 include a self-inversion at the 3′ end of *GPC5* (Supplementary Figs. 8c, d and 10e). A scaffold from the Irys-based reconstruction indicated a secondary reconstruction could be joined with the *BCR-ABL1* reconstruction (Supplementary Fig. 8d; overlap of segment 20). From the AR reconstructions of the *BCR-ABL1* amplicon and the co-existence of *BCR*, *ABL1*, and *GPC5* in overlapping locations, as shown by FISH (Fig. 3c 'Zoom'), AR enabled us to hypothesize a potential sequence of events by which the fCNA formed. The AR reconstructions support the formation of the *BCR-ABL1* translocation (Supplementary Fig. 10g;i–ii) followed by incorporation of chr13 regions (Supplementary Fig. 10g;iii-iv), which subsequently undergo rearrangement (Supplementary Fig. 10g;v), and ultimately a series of inverted repeats, possibly mediated through dicentrism (Supplementary Fig. 10g;vi). These results are consistent with previous reports using cytogenetic approaches to observe the presence of additional chromosomal segments besides chr9 and chr22 involved in the Philadelphia chromosome[30,31].

**AR enabled the reconstruction of a BFB**. The BFB mechanism of genomic amplification involves the loss of telomeres and subsequent fusion of two sister chromatids[12,13]. In subsequent cellular division, the asymmetric breaking of the fused chromosome structure results in one daughter cell acquiring additional pieces of the previously fused chromosome. The structure of various BFBs have been analyzed using cytogenetic techniques[14] and by computational models that predict BFB presence from copy number counts[42,43]. Both methods are imprecise, to a degree, and may fail to capture the fine structure of the BFB or handle imprecise copy number counts and/or additional structural variants (SVs) inside the BFB. We deployed AR on the HCC827 lung cancer cell line where AA and cytogenetics suggested a chr7 BFB, though an unambiguous structure was not identifiable[5,6].

We observed a banded pattern of *EGFR* and CEP7 (a chr7 centromeric D7Z1 repeat) in a DNA FISH experiment on HCC827 cells, suggestive of a BFB mechanism (Fig. 4a). AA generated a breakpoint graph of a 4.2 Mbp amplified region of chr7 containing *EGFR* (Fig. 4b). The amplified BFB segments in the AA output ranged in size from 217 to 1176 kbp. AR enabled the reconstruction of 16 unique OM scaffolds which, when combined, enabled the reconstruction of the BFB structure (Fig. 4c, d). The five most informative single scaffolds ranged in size from 750 kbp to 2.3 Mbp, containing multiple junctions which validate the order and orientation of the BFB breakpoint graph segments, resulting in a 9.4 Mbp amplicon, hereafter referred to as a BFB repeat unit. The BFB repeat unit was amplified across the chromosome (Fig. 4a, e, f). AR also revealed a region outside the AA amplicon, near the centromere of chr7, which explained the observed *EGFR* and CEP7 repeat (F). In segment B, we observed a 600 bp deletion across the entire BFB repeat unit and an 11 kbp inversion. The latter is labeled throughout Fig. 4 with a black asterisk and only appears when segment B is duplicated and inverted, suggesting that the SV arose midway through the formation of the BFB. While some BFBs may result in double-minute amplicons[7], AR suggested, and FISH analysis confirmed that the HCC827 BFB does not contain a circular extrachromosomal version of the BFB cycle.

When the AR scaffolds were combined with the copy number data present in the breakpoint graph, we could manually identify a complete BFB structure consistent with the theoretical model of BFB formation[44]. A putative sequence of BFB cycles and additional structural variation results in the final BFB structure is shown in Fig. 4f (also Supplementary Fig. 11a, b). Without AR, the copy number information and the theoretical model together could not have reconstructed this BFB, as it contains heterogeneous interior structural variants. We further validated the BFB patterning in HCC827 cells with multi-FISH for segments A, C, and D from the BFB, using FISH (Fig. 4e, Supplementary Fig. 11c). Together, these results show the ability of AR to enable the resolution of a BFB-driven fCNA, even in the presence of additional structural heterogeneity.

In addition to the *EGFR*-bearing amplicon, AA detected five other amplicons containing *MYC* and *NCOA2*, among other oncogenes. The graphs were complex (Supplementary Fig. 12a) and in many cases AA did not identify discordant edges between distinctly amplified regions. Given the dearth of breakpoint edges, we combined the amplicon breakpoint graphs for all six HCC827 amplicons and ran AR on the combined graph, containing 555 segments. AR identified 206 contigs having alignments to one or more graph segments. AR reconstructed multiple possible scaffolds and captured overlapping subsets of amplicon regions from different graphs, suggestive of possible amplicon heterogeneity. One scaffold showed *NCOA2* located on a native region of chr8, while another showed *NCOA2* joined to *MYC* through a segment of chr21 (Supplementary Fig. 12b, c).

**Additional focal amplifications reconstructed by AR**. In breast cancer cell line T47D, where the AA breakpoint graph suggested amplification of a 634 kbp region, AR reconstructed a 430 kbp segmental tandem duplication, containing oncogene *GSE1* (Supplementary Fig. 13a, b). This highlighted the ability of AR to also reconstruct classes of ultra-large, albeit less-complex SVs.

In renal cancer cell line, CAKI-2, AA generated a breakpoint graph spanning 12.0 Mbp, joining regions from chr3 and chr12 (Supplementary Fig. 13c, d). Despite the lower overall copy number of this amplicon (~5), AR still reconstructed a 13.1 Mbp amplicon explaining 99.9% of the amplified genomic content in the AA-detected fCNA. Both amplicons for CAKI-2 and T47D appear to be intrachromosomal events given the AR results.

Across the focal amplifications we studied in seven cancer cell lines, we report 64 individual amplified breakpoints detected by both AA and validated by AR (Supplementary Data 3). We also report a summary of all reconstruction findings and a list of reconstructed paths in Supplementary Data 4. Taken together, our data demonstrate the power of AR to combine NGS and OM data to elucidate a variety of complex fCNAs commonly found in cancer—enabling a deeper understanding of the fundamental

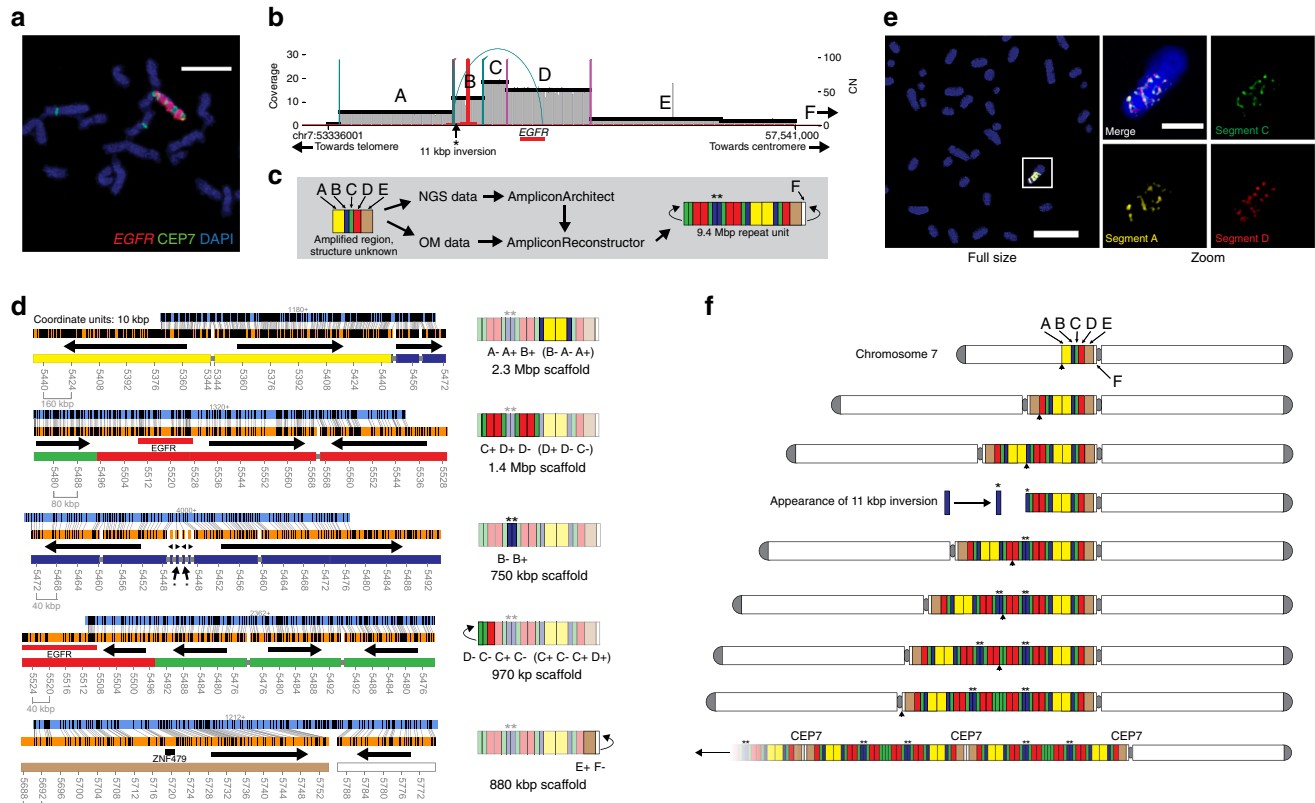

**Fig. 4 Reconstruction of Breakage–Fusion–Bridge. a** FISH confocal microscopy of DAPI-stained metaphase chromosomes in HCC827 showing multiple distinct bands of *EGFR* and CEP7 (chr7 centromeric repeat probe). Scale bar indicates 6 μm. **b** AA-generated breakpoint graph for amplified *EGFR* region in HCC827. Asterisk ('*') symbol indicates presence of 11 kbp inversion at 5′ end of segment B. **c** Workflow for analysis of amplified *EGFR* region in HCC827 to reveal BFB repeat unit structure. Amplified intervals detected by AA are labeled A–E and are colored yellow, blue, green, red, and brown, respectively. Segment F indicates a region identified by AR but not AA. **d** Visualization of the AR-generated scaffolds (left column) and cartoon illustration of reconstructed region(s) of the BFB (right column), including segment sequence. Black arrows in the scaffold column indicate segment directionality. **e** Multi-FISH for BFB segments using super-resolution confocal microscopy on DAPI-stained metaphase chromosomes in HCC827. FISH probes used for segments A, C, and D were RP11-64M3, RP11-117I14, and *EGFR*, respectively. Scale for full size image indicates 11 μm. Scale bar for zoomed images indicates 3 μm. Brightness was decreased using ImageJ between full size and zoomed images. **f** Theoretical model of formation for HCC827 *EGFR* BFB. Each row indicates a prefix inversion and duplication characteristic of BFB, alongside other SVs. Black arrowheads beneath the intermediate step in each row indicates the breakpoint of the BFB chromosome. The bottom row shows multiple duplications of the BFB unit along with a pericentromeric region of chromosome 7.

mechanisms that give rise to fCNAs and promote cancer pathogenesis.

**Integration points of focal amplifications.** Low-frequency breakpoint edges, such as the ones indicating integration points may not appear in the NGS breakpoint graph and may not be seen in assembled OM contigs. Using alignments of single-molecule optical maps, generated by the Bionano RefAligner molecule alignment pipeline, we gathered molecules with split alignments joining a partial alignment inside the amplicon region with a partial alignment outside the amplicon region. For H460, K562, CAKI-2, HCC827, and T47D, OM coverage was deep (>100, Supplementary Data 1) allowing us to cluster split alignments into OM-derived breakpoint clusters suggestive of low-frequency integration points. Requiring that each breakpoint cluster have 10 or more molecules suggesting the same junction (within 25 kbp on either side), we identified four integration point candidates (Supplementary Data 3).

Visualized with MapOptics[45], H460 showed a single integration point between amplicon region chr8:129410000 and non-amplicon region chr12:7660000 (Supplementary Fig. 14a). K562 showed two integration points. The first joined amplicon

region chr13:81120000 and non-amplicon region chr1:142890000 (Supplementary Fig. 14b). The second joined amplicon region chr13:93260000 and non-amplicon region chr1:142890000 (Supplementary Fig. 14c). The proximity of these two integration points suggests a left and right boundary for the integration of the K562 *BCR-ABL1* amplicon. CAKI-2 showed one integration point joining amplicon region chr12:88300000 and non-amplicon region chr6:168380000 (Supplementary Fig. 14d).

HCC827 and T47D did not show any such integration points with 10+ molecules of support, which is consistent with the finding that these were chromosomally derived focal amplifications (BFB and segmental tandem duplication, respectively).

**AR provides a reconstruction improvement over AA.** AA can identify some putative paths and cycles in the breakpoint graph only using NGS data. We demonstrated that for complex amplicons, AR provides an improvement to the fraction of the amplified genomic segments in the heaviest reconstruction path or cycle compared to the heaviest path or cycle generated by AA (Supplementary Fig. 15a). OM data may suggest additional amplicon junctions not observed in NGS. The segment junctions observed in the AR output was equal to (GBM39, T47D) or larger

than (CAKI-2, H460, HCC827, HK301, and K562) the number of junctions suggested by the AA breakpoint graph alone (Supplementary Fig. 15b).

Dixon et al.[21] used an integrative approach to detect structural variation and associated breakpoints using a combination of NGS, OM data and other sequencing modalities. We identified four cell lines shared between our studies for which Dixon et al. reported breakpoints identified by their integrative approach. We observed that in regions analyzed by AR, more breakpoints were detected with AR than with the integrative approach, though there were some breakpoints indicated by Dixon et al. which were not observed by AR (Supplementary Fig. 15c). In those cases, the majority of breakpoints not observed by AR joined amplicon regions to regions outside the amplicon (CAKI-2: 2 of 3 non-AR breakpoints, H460: 1 of 1 non-AR breakpoints, K562: 11 of 16 non-AR breakpoints). In H460, the one breakpoint not observed by AR was the integration point we later detected, suggesting that these are lower frequency breakpoints perhaps related to integration.

## Discussion

Revealing the architecture of fCNAs, particularly at a large scale, is critical to understanding the functional consequences. For instance, rearrangements present in fCNAs frequently increase oncogene copy number[46], disrupt gene structure[47], and lead to dysregulation of chromatin[17–19]. Accurate reconstruction of fCNA architecture can provide insights into the mechanisms of formation, leading to an improved understanding of the biological consequences of fCNA that would not be available solely from methods characterizing individual breakpoints. AR does not yet automatically produce a prediction of the biological mechanism of amplification. Thus, the AR reconstructions still require some manual interpretation based on the visualized results.

While previous methods have characterized complex structural variation using both OM and NGS data[21,48], these methods have typically focused on individual variants and breakpoints[42]. OM tends to detect larger SVs than NGS alone and is less affected by mapping issues on low-complexity breakpoints[21,24]. We have demonstrated that NGS data, when incorporated with OM can be used to resolve fine-mapped breakpoints suggested by OM. Indeed, some of the individual junctions reported by AR in these cell lines were already known[21] (Supplementary Fig. 15c). However, AR represents a robust, comprehensive algorithmic approach to reconstructing the fine-scale and large-scale structure of an fCNA through the propagation of NGS-derived breakpoint information into larger scaffolds.

Many variables affect the ability to resolve fCNA. Importantly, the complexity and structure of the fCNA, as well as the length of the reads or genome maps. Further compounding the difficulty of fCNA reconstruction, we note that different sequencing modalities do not overlap perfectly in the breakpoints they detect[21,24]. Based on our findings, we suggest the resolution of chained breakpoints should be spanned by long-range sequencing data with length sufficient to anchor the chain on both ends. We attribute much of the success of AR for resolving fCNAs to the long molecule length (244 kbp median) in comparison with the length of amplified genomic segments in the breakpoint graph (100 kbp median).

The paths reconstructed by AR represent possible reconstructions of an fCNA, and may contain multiple similar explanations for the fCNA architecture. This may be in part due to amplicon heterogeneity, limitations of the optical map assembly process, or errors in linking scaffolds across overlapping graph segments. Despite the integrative nature of AR's inputs, breakpoints may still be missed in the amplicon. One traditionally difficult case to reconstruct involves nested duplication of genomic segments inside an amplicon. Unless a significant fraction of reads or genomic maps have a length greater than the duplicated element, the duplication status may not always be accurately resolved, leading to ambiguity. Multiple tandem duplications can also give rise to a cyclic breakpoint graph structure. However, in that case the same breakpoint would be reused repeatedly, and evidence points against that possibility[5,6,46]. Instead, ecDNA-derived mechanisms provide a simpler and arguably more correct interpretation of cyclic graph structures, as validated by cytogenetics and comparison to Circle-seq experiments[46,47].

Genomic structural heterogeneity is problematic for any genome reconstruction, including focal amplifications. Despite the change in topology between linear HSR-like and circular ecDNA fCNAs, the breakpoint graphs between both circular and linear forms of the same samples are highly similar[6], suggesting ecDNA genomic structure is often not altered during reintegration. While we analyzed data from cancer cell lines, sequencing data collected from patients may introduce more sources of complex genomic structural heterogeneity. Assembled OM contigs may fail to capture rare instances of structural heterogeneity in the genome. However, previous results suggest that focal amplifications conferring a fitness advantage to cancer cells are clonally amplified[5,49], allowing for accurate reconstruction of the dominant structure.

AR produced a high-confidence reconstruction of the K562 BCR-ABL1 focal amplification yet copy number variance in this amplicon not explained by AR may be due to structural heterogeneity across the many copies of the amplicon. Additional copy number changes in K562 near BCR and ABL1 which are not directly explained by the amplicon through edges identified by NGS reveal limitations to our current method, or possible inaccuracies. Such cases may indicate additional amplicon segments outside the regions reconstructed by AR, suggesting that the true amplicon structure may extend beyond the regions we have captured. Despite the presence of the AR-supported and FISH-validated HSR-like status of the BCR-ABL1 translocation in K562, there does not exist a completely validated model that explains the increased copy number of BCR-ABL1 in one single location. We cannot rule out the possibility that the BCR-ABL1 amplification in K562 is mediated through an ecDNA stage[50], given the transient nature of the emergence and retreat of ecDNA[15] and the highly rearranged genomic landscape surrounding BCR-ABL1.

We have not yet adapted AR to accept data generated by other long-range sequencing modalities, breakpoint graphs generated by other tools or to accept breakpoint graphs derived from non-amplified rearrangements. Recent advances in other long-range sequencing technologies[51] highlight the need to adapt the AR algorithm. With modified protocols, nanopore reads may routinely surpass 150 kbp in length—sufficient to frequently chain multiple breakpoints in fCNA. We plan to address this in future methods development. Other sequencing modalities involving NGS with modified sample preparation, such techniques based on Hi–C and linked reads, have shown the ability to reveal additional genomic breakpoints without an additional sequencing instrument[21,24]. While de novo breakpoint graph construction is not a part of the AR algorithm, we acknowledge that such techniques would be valuable to adapt for breakpoint graph generation.

Methods to accurately characterize fCNAs will enable better classifications of cancer subtypes and their associated prognoses. The accurate, multi-megabase scale, complex fCNAs reconstructed by AR not only describe fine structural features of fCNA architecture, but also reveal mechanistic signatures of fCNA formation, allowing for future interrogation of the relationship between fCNA architecture and the biological consequences of their structure.

## Methods

**Cell culture**. NCI-H460, K562, and HCC827 cells were obtained from ATCC and cultured in RPMI-1640 media supplemented with 10% FBS. HK301 cells were cultured as neural spheres in DMEM/F12 media supplemented with B27, EGF (20 ng/ml), FGF (20 ng/ml), and heparin (1 ug/ml). All cells were incubated under standard conditions.

**Metaphase chromosome spreads**. Metaphase cells were enriched by treating cells with Karyomax (Gibco) at a final concentration of 0.1 µg ml$^{-1}$. Cells were collected, washed in PBS, and resuspended in 75 mM KCl for ~15 min at 37 °C. Cells were fixed by addition of an equal volume of Carnoy's fixative (3:1 methanol:glacial acetic acid). Cells were washed three additional times in Carnoy's fixative and dropped onto humidified glass slides.

**FISH**. Metaphase spreads were equilibrated in 2x SSC (30 mM sodium citrate, 300 mM NaCl, pH 7) for ~5 min. They were dehydrated using successive washes of 75, 85, and 100% ethanol for 2 min each and allowed to dry. FISH probes were diluted in hybridization buffer (Empire Genomics) and added to metaphase spreads on slides, along with 22-mm$^2$ coverslips. Samples were denatured at 70–75 °C for 30 s–2 min. Probe hybridization was performed at 37 °C for around 3 h or overnight in a humid and dark chamber. Samples were washed successively in 0.4x SSC and 2x SSC with 0.1% Tween-20. Samples were incubated with DAPI (0.1 µg ml$^{-1}$ in 2x SSC) for 10 min, then washed with 2x SSC and briefly rinsed with H$_2$O. Samples were mounted with Prolong Gold, #1.5 coverslips, and sealed with nail polish.

**Microscopy**. Confocal microscopy was performed on a Leica SP8 Confocal microscope with white light laser and Lightning deconvolution. Fluorescent microscope images were acquired using an Olympus BX43 microscope with a QiClick cooled camera. Images were subsequently analyzed in ImageJ[52] (using the Bio-Formats plugin[53]), to perform cropping, add scale bars and perform global adjustments to image brightness.

**Acquisition of WGS data**. We previously published[5,6] WGS data on SRA for six of the seven cancer cell lines (GBM39, NCI-H460, HCC827, HK301, K562, T47D) analyzed here. For CAKI-2, we used WGS data published by the Cancer Cell Line Encyclopedia on SRA. A list of SRA accession numbers used is available in Supplementary Data 1.

**Breakpoint graph generation**. WGS data was aligned to hg19 with BWA-MEM[54] (version 0.7.17-r1188, default parameters), sorted and PCR-duplicate filtered with SAMtools (version 0.1.19-96b5f2294a)[55], and the resulting alignments along with SNV calls produced by Freebayes[56] (version v1.3.1-17-gaa2ace8) were supplied as input to the Canvas[57] CNV caller (version 1.39.0.1598). The alignments and CNV seeds were filtered using AmpliconArchitect's amplified_intervals.py module. Seeds exceeding 40 kbp with copy number 5 were subsequently analyzed with AmpliconArchitect. AmpliconArchitect outputs a breakpoint graph encoding segmented CN calls and the discordant reads connecting the segments. We note that in most cases identical amplicon regions are identified when CNV caller ReadDepth[58] is used for seeding instead.

We standardized the breakpoint graph generation process into a workflow called PrepareAA, available on GitHub: https://github.com/jluebeck/PrepareAA. We used the default parameters specified by PrepareAA in this analysis. To produce in silico digestions of breakpoint graph segments into reference optical maps, we used the generate_cmap.py utility in AR. This method for in silico digestion can produce labeling patterns for the Bionano Saphyr DLE-1 labeling pattern, while many previous methods for in silico digestion do not.

**OM data generation**. High molecular weight (HMW) DNA was extracted from GBM39, HCC827, HK301, and K562 cells using the Bionano Prep Blood and Cell Culture DNA Isolation Kit (Bionano Genomics #80004), with minor modifications to recover good quality HMW gDNA. As detailed below, the Nick, Label, Repair, and Stain (NLRS) and Direct Label and Stain (DLS) reactions were carried out for the Bionano Irys and Saphyr platforms, respectively. To generate the Irys data, DNA was nicked using Nt.BspQI nicking endonuclease (NEB), followed by labeling, repairing, and staining, using the Bionano Prep NLRS DNA Labeling Kit (Bionano Genomics #80001) along with recommended NEB reagents. To generate the Saphyr data, DNA was labeled with DLE-1 enzyme, followed by proteinase digestion and a membrane clean-up step, using the Bionano Prep DLS DNA Labeling Kit (#80005). BspQI-labeled DNA was loaded onto the Irys Chip (Bionano Genomics #20249) and the run conditions were manually optimized on the Irys system (Bionano Genomics #30047) to ensure efficient DNA loading into the nanochannels. DLS-labeled DNA was loaded onto a Saphyr Chip (Bionano Genomics #20319), and run conditions were automatically optimized on the Saphyr system (Bionano Genomics #60239) using the Saphyr Instrument Control Software to maximize DNA loading. Raw images generated by Irys were processed into BNX files using the Bionano software AutoDetect[26]. Images from the Saphyr system were processed into digital BNX files via the Saphyr Instrument Control Software. For Irys data, molecules ≥150 kilobase pairs (kbp) were assembled into

consensus genome maps using the Bionano Assembler[29,30] (version 5122), using default parameters; for Saphyr data, molecules ≥150 kbp were assembled into maps using Bionano Access (version 1.2.1)[29]. Bionano Genomics separately provided Saphyr OM data for cell lines K562, T47D, NCI-H460, and CAKI-2. The methods by which OM data was generated for those four cell lines were previously published[21]. All Bionano software utilized alongside this study is available from the Bionano Genomics, Inc. website (https://bionanogenomics.com/support/software-downloads/) under the Bionano Genomics, Inc. software license (https://bionanogenomics.com/company/legal-notices/).

**Optical map contig alignments with SegAligner**. SegAligner uses a dynamic programming (DP) approach to optical map alignment, with a recursion similar to previously proposed DP algorithms for OM alignment[59,60]. SegAligner scores OM alignments in a manner accounting for collapsed pairs of labels in the assembled OM contig and uses an E-value approach to compute alignment significance as method of controlling false alignments. We define label collapse as the phenomenon where two nearby labels on an OM contig or map are measured as a single label due to limitations of imaging[61].

SegAligner supports alignment of in silico digested segments of the reference genome (including entire chromosomes of the reference genome) and assembled optical map contigs. SegAligner supports models of error for data from both the Bionano Irys and Bionano Saphyr instruments, and we parameterize our methods for them separately (Supplementary Data 5). SegAligner also supports multiple modes of alignment including semi-global, fitting, and overlap alignment.

To motivate the notion of an OM alignment, we first define the concept of an OM matching region. Similar to Valouev et al.[60], a matching region is defined as the region between and including two labels on a map. For example, $j$ and $i$ in Supplementary Fig. 1b constitute a matching region with size $j − i$ and one unmatched label in-between. The alignment score for two matching regions depends on the size discrepancy of the matching regions and the number of unmatched labels in each matching region.

We define the following variables:

- $b$ is a sorted list of real numbers corresponding to the positions of labels on the optical map contig in base-pair units.
- $x$ is a sorted list of real numbers corresponding to the positions of labels on a single in silico reference segment in base-pair units.
- $P$ is a matrix storing backtracking references.
- $U$ is a set storing reference segment label to contig label pairings which have already been used in previous iterations of the alignment process.
- $d$ is the width of the band to consider for a banded alignment (default 6).
- $M$ is a map which relates each label, $j$ on a genomic segment, $x$, to the estimated probabilities for the left neighbor and right neighbor of $j$, that $j$ and a neighbor would be observed as a single label (i.e., label collapse).

Next, we define $S[j][q]$ as the best score of aligning a subsequence of the first $j$ labels on $b$ with a subsequence of the first $q$ labels on segment $x$, where $j$ and $q$ are included in the subsequences. Given two labels on the assembled contig $i, j$, and two labels on the reference genome segment $p, q$ where $i < j$, and $p < q$, The DP recurrence used by SegAligner is

$$S[j][q] = \max_{\substack{\max(0, j-d) \le i < j \\ \max(0, q-d) \le p < q}} \{S[i][p] + \text{Score}(i,j,p,q)\}, \tag{1}$$

where the function Score is the SegAligner scoring function for two OM matching regions.

The function Score is defined in Algorithm 1 and contains four main terms. First, $f_n$ which is defined as the number of potentially unmatched contig labels between $i$ and $j$ scaled by the missing label score, $c$. Second, $e_{ref}$ is the number of potentially unmatched reference labels between $p$ and $q$, after accounting for labels which are too close together to be measured distinctly. Third, $f_p$ is the number of potentially unmatched reference labels scaled by the missing label score. Last, $\Delta$ measures the absolute difference in length between $j − i$ and $q − p$, which is scaled non-linearly ($k$). Together these penalty terms are combined and subtracted from a base matching score, $2c$. Parameters $c$ and $k$ in this model were identified through a coarse grid search using data where correct OM contig-reference alignments were already known.

**Algorithm 1** SegAligner scoring function

```
function Score(b, x, i, j, p, q, M):
    f_n = c * (j - (i + 1))
    e_ref = M(p, q)
    f_p = c * e_ref
    Δ = (abs((b[j] - b[i]) - (x[q] - x[p])))^k
    return 2c - (f_n + f_p + Δ)
```

A backtracking matrix, $P$ is used to record the decision made in filling each cell $S[j][q]$. The DP Algorithm has complexity $O(mnd^2)$ where $m = |b|$, $n = |x|$ and $d$ is the width of the band. Backtracking is performed in $O(m)$ steps by backtracking through the coordinates stored in $P$. We find a most-likely path by initializing the backtracking at $\text{argmax}_{j,q} S[j][q]$ or $S[|b| − 1][|x| − 1]$ for fitting alignment. Values

used to parameterize the scoring function and label collapse map generation function given below are provided in Supplementary Data 5.

As multiple regions of a long OM query might match similar regions of the reference, we extend the DP by masking out the best alignment path from the DP scoring matrix and recomputing the next best alignment. SegAligner uses a set ($U$) to keep track of the pairings of segment labels ($q$) and reference labels ($j$) which form each significant high-scoring alignment. After a highest scoring alignment is found, the label pairings ($j$, $q$) are added to $U$. Subsequent alignments of that segment cannot reuse any pairings in $U$. This limits the creation of many nearly identical local alignments which differ by small indels, only one of which (the highest scoring) is useful from a practical standpoint. We also placed a threshold on the number of times a single segment can be aligned to a single contig, so that low-complexity segments do not cause the aligner to stall (default 12).

Labels within ~2000 bp on an OM molecule may be read as a single label due to limitations of imaging, with increasing probability for smaller label-to-label intervals (Supplementary Fig. 1c). SegAligner captures that behavior in its scoring method, by precomputing the number of expected labels appearing in a collapsed label-set, given the reference.

To compute probabilities of label collapse, we assume a model in which the probability that a label at position $r$ has merged with its right neighbor at position $s$ is given by

$$P(r \rightarrow s) = \min\left(1, \left(\frac{(s-r)^t}{w^t}\right)\right). \tag{2}$$

The map $M$, encoding the expected number of uncollapsed labels between two points on an in silico reference segment, is generated iteratively, by evaluating the following sum. $M(p,q)$ represents the sum of probabilities for each label between, but not including $p$ and $q$ that the label has collapsed with a neighbor. The sum of probabilities for [0,1] binary random variables to equal 1 naturally gives the expected value of the sum of the binary random variables.

$$M(p,q) =$$
$$\begin{cases} \sum_{p<k<q}\left(1-\min\left(1,\frac{(x[k]-x[k-1])^t}{w^t}\right)\right)\left(1-\min\left(1,\frac{(x[k+1]-x[k])^t}{w^t}\right)\right), & \text{if } x[q]-x[p] \geq \eta \\ 0, & \text{otherwise} \end{cases}. \tag{3}$$

A genomic segment may appear multiple times in an optical map contig. Values for $w$, $t$, and $\eta$ are parameterized separately depending on the Bionano instrument used (Supplementary Data 5). We selected default parameters separately for Bionano Irys and Bionano Saphyr instruments based on the tendency for the newer Saphyr instrument to have less directional uncertainty and a lower rate of label collapse. We selected default values for each instrument through a coarse grid-search strategy and manual examination of data with known alignments.

**Identifying significant high-scoring alignments.** To compute statistically significant alignments, SegAligner uses a strategy similar to BLAST[62]. For each reference segment, $r$, SegAligner constructs a distribution of alignment scores representing the highest scoring alignments of $r$ to all contigs (Supplementary Fig. 1b). As this distribution may contain true alignments between $r$ and one or more contigs, violating the random pairing assumption of the E-value model, SegAligner removes the highest 25 values from the distribution. From the remaining distribution of scores, we define a set of high-scoring segment pairs (HSPs) which are the distribution of scores from the 85th percentile and up, from which SegAligner estimates parameters in the E-value model. We note that this region of the HSP scoring distribution tends to behave linearly (Supplementary Fig. 1c), allowing for a linear regression approach to parameter estimation.

SegAligner assigns an empirical E-value for each element in the sorted distribution of HSP alignment scores based on its rank (highest scoring having E-value 1). SegAligner then performs a local linear regression to estimate unknown variables in the E-value model. Generally, the E-value model is given by

$$E = Kmn_r e^{-\lambda S} \tag{4}$$

which implies

$$\log(E) = \log(Kmn_r) - \lambda S \tag{5}$$

where $m$ is the size of the combined collection of contig labels, $n_r$ is the number of labels on the reference segment, and $S$ is the alignment score. As $K$ and $\lambda$ are unknown and represent the intercept and slope, respectively, SegAligner determines them from the empirical distribution of scores and E values using linear regression.

With all parameters known, the number of random high-scoring alignments, $a$, with score $\geq S$ is given by a Poisson distribution

$$P(a) = \frac{e^{-E}E^a}{a!}. \tag{6}$$

This implies that finding at least one HSP for a given value of $E$ is

$$P = 1 - e^{-E}. \tag{7}$$

Thus, the score-cutoff $S_r^*$ corresponding to a given probability, $P$, for segment $r$, is

$$S_r^* = \frac{-\log\left(-\frac{\log(1-P)}{Kmn_r}\right)}{\lambda} \tag{8}$$

SegAligner assigns to each reference segment a score which corresponds to the $p$-value cutoff for alignment significance. Default $p$-values are $10^{-4}$ for semi-global alignment, $10^{-6}$ for overlapping alignment, and $10^{-9}$ for detection of new genomic reference segments aligning to contigs where the reference segment is not specified in the provided breakpoint graph segments (detection mode). The need for different $p$-value thresholds between the different modes of alignment is based on the different sizes of the search spaces possible in the different modes. Searching for alignments between contig and entire reference is the largest search space to consider and thus it gets to smallest $p$-value threshold in order to stringently control false discovery. The default $p$-values for each mode were assigned based on empirical testing of OM data with known alignments. SegAligner also computes the mean and median of segment-contig label pair alignment scores for each alignment exceeding the significance thresholds. Statistically significant scoring alignments failing mean and median thresholds (Supplementary Data 2) are filtered out. By default, AR attempts to align graph segments with at least 10 (Irys) or 12 (Saphyr) labels in the segment. The need for different length thresholds is motivated by the different in labeling density between Irys and Saphyr. However, the fitting mode of alignment only requires two endpoint labels, and so it is used in the path imputation step in AR.

**Identifying unaligned amplicon contig regions with AR.** AR coordinates the alignment of in silico digested breakpoint graph segments to optical map contigs using SegAligner (Fig. 1b). Alternately, AR can take as input XMAP-formatted alignments produced by other alignment tools. If OM contigs with alignments to graph segments contain unaligned regions with between 20 and 500 unmatched labels, and 200 kbp to 5 Mbp in length, those regions are extracted and searched against the reference genome. The module ARAlignDetect calls SegAligner in the detection mode, which then aligns the extracted unaligned region of the contig(s) to the specified reference genome. If significant alignments are found between unaligned regions of the contig and chromosomal segments in the reference, those segments are extracted, and their identity is added to the collection breakpoint graph segments. Finally, a new breakpoint graph is output containing the newly detected segments.

**Reconstructing amplicon paths with AR.** Optical map alignments of segments with contigs are converted into a scaffold, which we define as a collection of alignments where the genomic distance between each pair of alignment endpoints is known. AR represents the scaffolded alignments as a directed acyclic graph (DAG), where the nodes are an abstract representation of each OM alignment. Directed edges connect adjacent alignment endpoints. Overlapping alignments are connected by special directed edges referred to as forbidden edges (Fig. 1h). Two nodes are only connected by a non-forbidden edge if the right endpoint of the source node has one or fewer labels of overlap with the left endpoint of the destination node. Each contig with at least one alignment to a graph segment will comprise an individual scaffold.

**Imputing paths in the scaffold with AR.** Some segments in the breakpoint graph may be too short to be uniquely aligned to an OM contig. AR attempts to impute corrected paths in the scaffold using the structure of the breakpoint graph. For every non-forbidden edge in the scaffold graph with a gap size <400 kbp, AR identifies breakpoint graph nodes corresponding to the source and destination endpoints, which we will denote as $s$, and $t$. AR then uses a constrained depth-first search (DFS) strategy to identify paths in the breakpoint graph between $s$ and $t$. Finding all possible paths between two nodes may produce infinitely many solutions should a cycle exist between the two nodes, so the recursion is constrained to terminate if certain conditions are reached. The constraints used in the search procedure are:

1. The multiplicity of the segments in the candidate path must always remain less than or equal to the copy number of the segment as specified in the breakpoint graph.
2. If a candidate path reaches the destination vertex, its length in base-pair units must not be more than $\min(25000, 10000L_p)$ shorter than the distance between the source and destination vertices as expected given the scaffold backbone, where $L_p$ is the length of the path in number of segments.
3. During path construction, the length of a candidate path must not exceed $\min(25000, 10000L_p)$ beyond the of the expected distance given the scaffold backbone.
4. The number of valid candidate paths connecting source to destination must not exceed $2^{10}$.
5. The path may not form a trivial cycle from ultra-short breakpoint graph segments <100-bp long. Such cycles appearing in an NGS-derived breakpoint graph we assumed to be erroneous or artifactual.

As constraint #4 may cause failure of the DFS whereby a tractable number of paths is not found, AR implements a constrained BFS search as a fallback option, which is used when the DFS fails for that reason. By parsimony, shorter paths between two nodes are more likely to be correct, thus AR applies the same set of criteria for the BFS search, with the threshold in constraint #4 increased to $2^{16}$.

All valid candidate imputation paths discovered by AR are scored by a fitting alignment procedure using SegAligner. To score a candidate path, the ordered path segments, as well as the first and last labels on the source and destination endpoints, are converted to a compound CMAP composed of the concatenated CMAPs of the individual segments. A fitting alignment is performed between the compound CMAP and the region of the contig between the alignment endpoints, using SegAligner. The path with the alignment score which most improves the junction score is kept. If no valid candidate path improves the score of the junction, it remains unimputed. The scaffold is then updated to contain the imputed breakpoint graph path.

**Identifying linked scaffold paths with AR**. Given the collection of scaffold DAGs, AR first searches for paths in the individual DAGs which represent heaviest paths in the scaffold DAG, where the weight of a path is the sum of the lengths of its segments in base pairs. AR stores the heaviest path(s) for each scaffold prior to performing scaffold linking.

AR leverages the two independent sources of information encoded in the breakpoint graph and OM contigs to link individual scaffolds. As the breakpoint graph segments are not detected to contain interior breakpoints, two endpoint alignments of the same breakpoint graph segment may be linked across two contigs. AR searches for prefix paths and suffix paths in each DAG. From the collection of prefixes and suffixes, AR searches for overlap between scaffolds generated from different contigs. Given that a contig can be assembled in either direction, overlapping reverse oriented suffixes or prefixes can also be matched. AR exhaustively finds sub-paths hitting either end of a scaffold DAG, which have overlap with other endpoint sub-paths, where the endpoint sequence of the scaffold may be assembled in either direction.

**Finding reconstructions in the linked scaffold graph**. Given the graph of linked scaffolds, AR searches for paths in the graph which conform to the ratio of estimated copy numbers between the graph's amplified segments. AR starts by searching for all paths in the graph which begin at endpoint nodes in the individual scaffolds. AR then uses a greedy approach to identify the longest unique paths which conform to the copy number restrictions. From the candidate paths, AR checks each path segment's multiplicity against the copy numbers encoded in the breakpoint graph in a ratio-dependent manner.

AR iterates over all the segment multiplicities in the reconstructed path, and at each multiplicity level determines the maximum estimated genomic copy number of path segments with that multiplicity. If a path segment has a multiplicity that is greater than the genomic copy number of that segment divided by the maximum copy number of all segments with multiplicities less than the given segment, then the path violates the copy number ratio check. AR allows each segment in the reconstructed path to exceed by 1 copy, the copy number expected, given the ratio between breakpoint graph copy numbers and segment multiplicity. If $n_p$ is the multiplicity of segment $n$ in the candidate path, $P$, and $n_g$ is the copy number of graph segment $n$ in the breakpoint graph, then $n_p$ must satisfy

$$n_p \leq \max\left(c, \frac{n_g}{m_g}\right) + 1, \forall n \in P \tag{9}$$

where

$$m_g = \max\left(i_g, \forall i \in P, i_p == c\right) \tag{10}$$

$$c \in \mathbb{Z} \tag{11}$$

$$n_p > c > 0. \tag{12}$$

If a candidate path passes the copy number ratio check, it undergoes a pairwise comparison with other paths passing this criterion, to check for path uniqueness. A path is unique if it does not represent a subsequence of a previously identified unique path. Furthermore, no rotation of the path sequence may be a subsequence of a previously identified unique path. AR assess subsequence paths by computing a longest common substring between a candidate path and a previously identified unique path (Algorithm 2). As the paths are first sorted by total alignment score prior to the iterative approach, this method is a greedy algorithm which prioritizes long, heavy paths as being more likely to be identified as unique non-subsequence paths. AR categorizes paths as being cyclic if the first and last scaffold graph node in the path are the same, and the path length is >2, as this distinguishes cyclic paths from paths which appear cyclic such as singleton paths or paths which represent segmental tandem duplications. Paths reported by AR are output in the AmpliconArchitect-cycles file format.

**Algorithm 2** Greedy filtering of subsequence paths

```
Function FilterSubsequencePaths(sorted_paths):
    kept = empty array
    for P in sorted_paths do:
        isSubsequence = False
        for J in kept do:
            for R in the set of all rotations of path P:
                if R is a subsequence of J then:
                    isSubsequence = True

        if not isSubsequence then:
            append P to kept
    return kept
```

**Simulation of amplicons to measure AR performance**. We used OMSim[63] (version 1.0) to simulate Bionano Irys OM data from the hg19 reference as well as from 85 non-trivial paths (i.e., not directly consistent with the reference genome) in AA-generated breakpoint graphs from 25 cancer samples and 20 de novo simulated ecDNA structures, including both cyclic and non-cyclic breakpoint graph paths (Supplementary Fig. 3). OM molecules were simulated at 40× baseline coverage for each chromosome arm in hg19. The combined hg19 maps from all arms were assembled into a set of OM contigs using Bionano Assembler (version 5122). A similar process was performed using high-confidence breakpoint graph paths, which were converted to FASTA format and used for map simulation. For each simulated path, molecules were simulated at a range of copy numbers, and simulated molecules from the chromosome arm(s) (downsampled to the appropriate CN) from which the path segments came were combined and de novo assembled into OM contigs with Bionano Assembler. The resulting contigs from each amplicon simulation were combined with the previously simulated reference contigs and used as input to AR. For combination sets of three amplicons from the same sample, a similar downsampling and combination strategy was used, where molecules from each of the three amplicon simulations was separately downsampled based on the copy number settings of the mixture then combined. As heterogeneous combinations of amplicons may occur at different ratios, we selected three sets of copy numbers for this combination simulation cases: 20-20-20, 20-15-10, and 20-2-2.

In the simulation of the 20 de novo circular amplicons, a simulated tumor reference was generated from hg19 using SCNVSim (version 1.3.1) and simulated amplicon structures were generated using ecSimulator (version 1.0, https://github.com/jluebeck/ecSimulator). OM molecules were generated at baseline 40× coverage and amplicon copy number of 20. The human papillomavirus-16 integration example was performed at the same coverage and copy numbers as the other simulated amplicons.

**Measuring AR simulation performance**. We computed the longest common substring (LCS) between the AR paths and the ground-truth path and considered only the path having the LCS between AR and AA paths when computing precision and recall. We define the LCS here using the identities of the breakpoint graph segments and their orientations. We pre-filtered some possible assembly error reflected in the paths by removing ends of reconstructed paths which were trivial reconstructions of the reference genome and which were not supported by the AA path. To measure the accuracy of AR-reconstructed paths against the ground-truth simulated paths, we developed a set of three measurements which were used in calculating performance and recall.

(1) Length (bp): Reports the length of a breakpoint graph path in base-pair units.
(2) Nsegs: Reports the length of a breakpoint graph path in terms of the number of graph segments (unbiased toward genomic length).
(3) Breakpoint: Reports the length of a breakpoint graph in terms of the number of breakpoint graph segment junctions in the path.

We define precision and recall as follows, where $M$ is the path measurement function (Length (bp), Nsegs, or Breakpoint), $LCS$ is the longest common substring function, $P_{AA}$ is the sequence of segments in the AA path, and $P_{AR}$ is the sequence of segments in the reconstructed AR path:

$$\text{Precision:} \quad \frac{M(\text{LCS}(P_{AA}, P_{AR}))}{M(P_{AR})} \tag{13}$$

$$\text{Recall:} \quad \frac{M(\text{LCS}(P_{AA}, P_{AR}))}{M(P_{AA})}. \tag{14}$$

To summarize the precision and recall metrics in a single value, we computed a mean F1 score across all the simulated amplicons for a given set of simulation

conditions as

$$\text{mean F1} = \frac{\sum_i \left( 2 \frac{\text{precision}_i * \text{recall}_i}{\text{precision}_i + \text{recall}_i} \right)}{n}. \tag{15}$$

**Reconstructed path visualizations**. We developed a visualization utility, CycleViz (https://github.com/jluebeck/CycleViz), which produces circular and linear visualizations of AR or AA-reconstructed amplicons (Supplementary Fig. 2a, b), to create topologically correct visualizations of AR reconstructions. CycleViz accepts inputs including the path files reported by AR (in the AA-cycles format) as well as the path OM alignment files (optional) and produces visualizations which show the reconstructed path, in silico digestion of the path segments and the alignments of the digested segments with assembled OM contigs. For circular and linear visualizations, CycleViz places path segments in the visualization based on the length of the segments and their position in the path. For circular visualization layouts, the relative positions are converted to polar coordinates and a circular layout is formed. We also developed a web-based visualization utility, ScaffoldGraphViewer, for visualizing JSON-encoded scaffold graphs generated by AR using CytoscapeJS[64] (Supplementary Fig. 2c). The ScaffoldGraphViewer web utility can be accessed at https://jluebeck.github.io/ScaffoldGraphViewer/.

**Statistics and reproducibility**. We have provided instructions for using AR and commands for generating the results from the GBM39 cell line as an example on the AR GitHub page. The python package SciPy[65] was used to perform statistical tests in this study. All FISH experiments involved the analysis of at least three independent images and representative results are shown in the figures present in the study.

**Reporting summary**. Further information on research design is available in the Nature Research Reporting Summary linked to this article.

## Data availability
The AA-generated breakpoint graphs for the cell lines in this study are available on figshare with the identifier 10.6084/m9.figshare.11691798 (https://figshare.com/articles/AA_breakpoint_graphs/11691798). The FISH data that support the findings of this study are available on figshare with the identifier 10.6084/m9.figshare.11691774 (https://figshare.com/articles/FISH/11691774). Assembled Bionano contigs that support the findings of this study have been deposited in GenBank with Bioproject codes PRJNA602907 and PRJNA506071. The SRA experiment IDs associated with WGS data for cell lines appearing in this paper are SRX2666689 (CAKI-2), SRX2006441 (GBM39), SRX2006457 (H460), SRX3769666 & SRX3769671 (HCC827), SRX2006453 (HK301), SRX2006506 (K562), and SRX2006468 (T47D). The remaining data are available in the Article, Supplementary Information or available from the authors upon reasonable request. Source data are provided with this paper.

## Code availability
The following tools are available online: PrepareAA: https://github.com/jluebeck/PrepareAA. AmpliconReconstructor (& SegAligner): https://github.com/jluebeck/AmpliconReconstructor. CycleViz: https://github.com/jluebeck/CycleViz. ScaffoldGraphViewer: https://github.com/jluebeck/ScaffoldGraphViewer.

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

## Acknowledgements

The authors thank members of the Bafna and Mischel labs for their input, Dr. Marcy Erb (UCSD SOM Microscopy Core) for advice on FISH experiments, the Ecker Lab at the Salk Institute for use of the Bionano Irys optical mapping instrument, and Bionano Genomics, Inc. (Alex Hastie, Ernest Lam, Andy Pang, Jian Wang, Heather Mashhoodi and others) for supplying data and providing input on this project. Work in the Bafna group was supported in part by grants from the NIH (GM114362, RR24851). Work in the Law laboratory was supported by a Salk Innovation Grant and by the Rita Allen Foundation Scholars Program and the Hearst Foundation. C.C. was supported by a Postdoctoral Fellowship from the Glenn Center for Research on Aging at the Salk Institute and from a Salk Women and Science award. Work involving microscopy was supported by the UCSD Microscopy Core (NINDS NS047101) grant.

## Author contributions

J.L., V.B., and P.S.M conceived the work and designed the study. J.L. and V.B. developed the AmpliconReconstructor algorithm and software. C.C. and D.A.P. generated Bionano OM data and provided technical advice. J.L., S.R.D., and V.B. conceived and conducted the simulation study. J.T.L. and K.M.T. conducted FISH and microscopy experiments and provided technical advice. V.B., P.S.M., and J.A.L. supervised all experiments. V.D., C.Z., and U.R. performed computational analysis and provided technical advice. J.L., C.C., J.T.L., K.M.T., V.D., D.A.P., J.A.L., P.S.M., and V.B. wrote the paper.

## Competing interests

The authors declare the following competing financial interests. P.S.M. and V.B. are co-founders of Boundless Bio, Inc. (BB), and serve as consultants. K.T. is currently employed by and receives income from BB. V.B. is a co-founder, and has equity interest in Digital Proteomics, LLC, and receives income from Digital Proteomics (DP). D.A.P. is employed by and receives income from Bionano Genomics, Inc. The terms of this arrangement have been reviewed and approved by the University of California, San Diego in accordance with its conflict of interest policies. The rest of the authors declare no financial competing interests. The authors declare no competing non-financial interests.
