## [Peer Review File · Nature Communications]

Reviewers' comments:

Reviewer #1 (Remarks to the Author): Expert in computational biology and structural variants

Luebeck et al present Amplicon Reconstructor (AR), a tool for integrating NGS and optical mapping (OM) for the study of complex amplicons in cancer. The tool integrates NGS-derived breakpoint graph with OM scaffolds to infer end-to-end reconstructions of amplicons. AR includes "SegAligner" which realigns OM maps (or molecules?) to breakpoint graphs, which are then converted to DAGs by a "scaffolding module", from which paths are inferred (via a pathfinding module). Output from AR can be visualized using "CycleViz".

The authors use simulations to benchmark AR. They then apply the method to a set of cell lines with publicly available NGS WGS and OM data, of which 4 cell lines were profiled with OM as part of this study. Using AR, they propose the structure of complex amplicons resulting in the copy number gain of EGFR, MYC, BCR-ABL, including putative eccDNA / double minutes and what the authors characterize as the "first sequence based reconstruction of a breakage fusion bridge". They corroborate a subset of their findings with FISH.

The area of applying OM to the study of structural rearrangements in cancer is an important one, and in need of integrative algorithms like the authors propose. The method is novel and has interesting elements, including the realignment approach and a reasonable framework for integrating NGS with OM.

A few additional clarifications and analyses would make this a strong paper. In particular, it would be more productive for the field to acknowledge the residual ambiguities in amplicon structure inference for certain event classes as a function of OM molecule size. In addition, since exact amplicon structure inference likely can't be guaranteed in many cases of cancer loci, it may be better to gear the method towards answering biological questions that don't rely on exact inference (eg ecc vs integrated).

Major critiques:

* The paper / field would benefit from a better discussion of the ambiguities of amplicon reconstruction. Even with OM, the order of segments in a nested duplication may very likely be ambiguous, especially if the locus is much larger than an OM map. The authors acknowledge this ambiguity in the methods (mentioning that AR will only resolve events with fewer than 1024 possible paths) and a little in the main text (mostly referencing previous work / state of the field). However it would be useful to have a more textured discussion of this ambiguity and what cases may be particularly difficult for AR / OM.

A key ambiguity is distinguishing between reconstructions that place many duplicated segments on the same allele vs different alleles. The most extreme example is distinguishing an eccDNA vs chromosomally integrated segmented dup (see below); however, any complex nested (tandem or inverted) duplication pattern might suffer from the same issue. For example, distinguishing between an large eccDNA with many tandem copies of a region vs a collection of small eccDNA's each with a

single copy of a duplication. Can the authors add a discussion of these issues and relate them to the parameters of SV nesting / molecule size. Simulation could help inform this intuition.

* Related to above, a key (biological) question in resolving amplicon structure is whether these structures are circular / extrachromosomal vs chromosomal / integrated / HSRs. A "cycle" on the NGS breakpoint graph can be equivalently interpreted as a tandem segmental dup or a eccDNA. i.e. these patterns will look identical on NGS WGS. Indeed, in practice an (arbitrary, heuristic) copy number threshold is used to distinguish between (lower copy) tandem dups and (very high copy) DMs. OM can potentially distinguish between these two possibilities since the OM maps are on the scale that they may be able to unambiguously place a (reasonably small) amplicon on a chromosome. Do the authors claim that their method is able to make this eccDNA vs integrated seg dup distinction reliably? How confidently can the authors make this assessment? Ideally this would be shown with simulations where the simulated ground truth is eccDNA vs integrated. Such a simulation, if working properly would fail with NGS WGS short reads and would (for a certain event scales) succeed in resolving amplicons with OM.

* The simulation seems a bit simple and biased towards the authors' existing tools. Firstly the simulation models molecules rather than whole chromosomes, and not clear whether it is diploid or matches the ploidy of real cancer genomes. Secondly, the simulated chromosomes are currently based on paths outputted from AA breakpoint graph, which are likely short and fragmented since they derived from NGS WGS? These graphs also likely do not have connectivity in repeat regions since the AA graphs would miss these SV's in the short read NGS data. Finally, it's unclear whether the AA-derived simulated rearrangements would have the sort of nested (tandem or inverted) duplication structures that would make these kind of reconstructions truly challenging in practice. It would be more convincing to use a simulation that rearranges a fully diploid genome end to end and explicitly models a BFB and complex double minute. This would also help determine when / how well AR can resolve essential features of amplicons (ie chromosomal vs ecc)

* Can the authors draw a better conceptual comparison / contrast of AR vs AA, including comparing results? I get it that AR incorporates OM data, but don't the tools otherwise have the same goal? It seems that the authors only utilize AA for breakpoint graph inference, but if I recall AA also generates paths. How do the AR vs AA paths compare? Similarly, AR integrates the NGS and OM data and presumably also should create a modified (improved) breakpoint graph, which better account for copy number changes (see below). How do the AR vs AA breakpoint graphs compare? These comparisons would help provide intuition as to the value added by OM.

e.g. Figure 2A shows a breakpoint graph with many apparent copy number changes that are not associated with a structural rearrangement. This is presumably the NGS WGS graph inferred by AA, but AR should improve this graph? How well does it improve the graph? Can such an AR-improved graph account for all the copy number changes with a rearrangement or are there still links missing?

Minor critiques:

* The paths that are inferred by AR presumably imply a copy number, and then each path has a multiplicity. When integrated together, does this copy number match the total copy number of the

region? It seems like the method only uses the total copy number as an upper bound? If so, then how do the authors explain the remaining copies and breaks in the region?

* how did the authors arrive at the reconstruction of the time sequence of events for the BFB amplicon. Was this done manually or is it a part of AR? If the latter can the authors reveal the method? Are there alternative reconstructions? Otherwise, it would be clearer to emphasize that this part of the analysis was manual and not an output of AR.

* where does the BFB "end"? Does it acquire a telomere of another chromosome or is a new telomere synthesized at one of its ends?

Reviewer #2 (Remarks to the Author): Expert in optical mapping and genome assembly

In this manuscript, Luebeck and colleagues describe a new computational approach for the reconstruction of amplicon architecture via the integration of NGS and optical mapping data (AmpliconReconstructor, AR). Their manuscript builds on previous work from this group which used WGS NGS data to generate accurate breakpoint graphs for copy number alterations (AmpliconArchitect, AA). Unambiguous reconstruction of these alterations have remained a challenge with short-read data alone, and here they integrate OM with NGS to produce such reconstructions and discuss their functional implications.

The authors are right to point out that such ambiguities are an important problem in the field, with long-standing issues in understanding copy number alterations and the genetic structures within. This is especially true with the recent wave of studies exploring extrachromosomal circular DNA and the implications of such discoveries at the level of both the genome and epigenome, as well as with older questions involving BFBs and translocations.

Their approach and algorithm appear to be adept at reconstructing fCNA architecture, even if currently limited to samples with both NGS and OM data. The manuscript is well written, and highlights both the advantages and caveats to their approach, which is appreciated. The manuscript could be further improved by addressing some of the following points.

Main comments:

1) Given the great interest in the field in computationally distinguishing between extrachromosomal and intrachromosomal amplifications, can the data/approaches here offer any insight into whether HSRs might be genetically joining the linear chromosomes to which they appear to be physically attached as observed in FISH? As the authors point out, the differences in topology between ecDNA and HSR-like fCNAs don't lend themselves to differences in the breakpoint graphs, but these breakpoint graphs are in fact derived from the focal amplifications mapped in AA, correct? If HSRs are at all directly fused to the linear genome (and they may not be), are there any 'edge' cases demonstrating something resembling an 'exit' from the fCNA to the associated linear chromosome in NGS or OM data?

2) As the authors point out, a previous effort integrated NGS (WGS and HiC) and OM data to study structural variants (Dixon et al, Nature Genetics 2018). The Dixon et al analysis of data from the renal cancer cell line CAKI-2 yielded a chr2-chr3 fusion (Fig 1) as well as a chr6-chr8 fusion (Fig 2), whereas AR results on CAKI-2 in this paper yielded a segment of chr3 joined to chr12 (Supplemental Fig. S11c,d). It's clear that different approaches will always have certain advantages and disadvantages, but can the authors explain the discrepancies here? Also, given the increasing abundance of HiC maps, might they be used in future iterations to aid the AA if not the AR steps in the current approach?

3) In the authors' 2019 paper (Deshpande et al), they use Amplicon Architect in an innovative way to find human-viral fusion segments in cancer genomes and explore their functions and potential origins. With the OM strategy, how might incorporation of these additional (non-human) reference sequences work with the AR approach?

4) The authors state that future iterations of AR may involve input from other long-read sequencing technologies. Perhaps outside the scope, but given that they've worked with PacBio data in the past (Deshpande et al 2019), can they comment on read lengths/accuracies that would make such efforts worthwhile for increasingly used platforms like PacBio and Nanopore?

5) Given that all of the data in this manuscript has been generated from cell lines, can the authors comment on challenges they expect to encounter when attempting to reconstruct amplicons in primary patient data with this approach?

Reviewer #3 (Remarks to the Author): Expert in optical mapping and genome assembly

Luebeck et al. "AmpliconReconstructor: Integrated analysis of NGS and optical mapping resolves the complex structures of focal amplifications in cancer." This is a well-written paper that describes an exciting new piece of software AR. AR can combine sequencing (WGS) and optical mapping (OM) data to accurately reconstruct the rearrangement events that commonly occur in various tumor lineages. Specific examples of various rearrangement and copy number amplification mechanisms were inferred and illustrated nicely. I believe that AR would be a useful software for researchers aiming for accuracy and resolution of such events. There are still some issues though as I went through this paper.

I think one of the immediate questions a reader might have – what is the added benefit of OM on top of WGS? The authors need to make it explicit during the Discussion. There have been many separate studies looking at WGS alone, or OM alone, what are they missing? AR makes heavy use of the breakpoint graph from WGS, does it somehow limit AR from the discovery of additional events that are difficult to identify using WGS (e.g. lack of coverage for certain regions, variation in mappability across the genome). There are some brief hints of this mentioned in the Results but not summarized systematically which I think would be highly valuable. Similarly, what are the added benefits of WGS+OM on top of OM alone?

Some in-depth analyses may be needed to look at the results from simulation studies to understand the common error modes. What are the characteristics of the false positive or false negative, or poor LCS predictions from AR? While this reviewer has little doubts that AR would be useful – it is important to inform the readers when the software fails and how they fail. Would supplying more input data (both the WGS and OM) help in those cases? The mixture experiment on heterogeneity is very nice. Can we learn what is the limit of capability of detection from the mixture experiment? For example, in terms of the minimum level of abundance or the number of clonal subpopulations in the sample.

In the Introduction, please introduce the key algorithmic challenge and what is the main algorithm and statistical models employed in tackling this challenge. Please mention any prior statistical modeling work on the properties of NGS-read breakpoint graph and optical mapping data. For example, the source of variation on segment size, missing labels, uncertainties in the direction and connection within the breakpoint graph, etc. to provide some theoretical ground for the AR method later in the paper.

There were a number of issues in the Methods section. Please check the formulas and pseudocode more carefully. While I could conceptually follow most of it, I had some issues going through.

Check DP recurrence equation, especially the sub for the max (line 578), I don't think I agreed with the current upper bounds for i and p .

Algorithm 2: explain what c and k are and whether they are fixed parameters or estimated from the data (and how). What is the rationale behind this scoring function? Specifically, what is the role of each of these terms? Explanations would help.

Poisson distribution (line 658), should it be a minus or equal sign? The formula also seems wrong for the exponential of e part. Should it be $-E$ instead of $-a$?

Some parametrization needs a little more explanation. Explain why different modes of alignment warrant different levels of P-value threshold, and how are these thresholds determined. Parameterization of Irys and Saphyr is different due to their properties (e.g. Fig. S1d) – Supplementary Table 5 contains at least 5 parameters that are different, as well as the minimum number of labels (line 671). What is the process to tune these parameters per platform, at least on a high level? This will be relevant as Bionano continues to upgrade the chemistry and specs, and for people trying to optimize those parameters for their own data.

Finally, as a sincere suggestion to improve software usability, please include visualization or succinct textual outputs on the front page of README of the repo to make it friendlier to the new users so they can know what they expect. Please also consider including a docker image that has everything installed and a copy of the test data that contain a full pipeline script end-to-end, including the command of PrepareAA to build the initial breakpoint graph.

Reviewer #1 (Remarks to the Author):

“The area of applying OM to the study of structural rearrangements in cancer is an important one, and in need of integrative algorithms like the authors propose. The method is novel and has interesting elements, including the realignment approach and a reasonable framework for integrating NGS with OM.”

We appreciate this comment from the reviewer.

1. “The paper / field would benefit from a better discussion of the ambiguities of amplicon reconstruction. Even with OM, the order of segments in a nested duplication may very likely be ambiguous, especially if the locus is much larger than an OM map. The authors acknowledge this ambiguity in the methods (mentioning that AR will only resolve events with fewer than 1024 possible paths) and a little in the main text (mostly referencing previous work / state of the field). However it would be useful to have a more textured discussion of this ambiguity and what cases may be particularly difficult for AR / OM.

“A key ambiguity is distinguishing between reconstructions that place many duplicated segments on the same allele vs different alleles. The most extreme example is distinguishing an eccDNA vs chromosomally integrated segmented dup (see below); however, any complex nested (tandem or inverted) duplication pattern might suffer from the same issue. For example, distinguishing between an large eccDNA with many tandem copies of a region vs a collection of small eccDNA's each with a single copy of a duplication. Can the authors add a discussion of these issues and relate them to the parameters of SV nesting / molecule size. Simulation could help inform this intuition.”

Response: We agree with the reviewer that a discussion of reconstruction ambiguity, and the ability or inability to distinguish between large duplications inside an amplicon are important to provide a complete characterization of the field, and this method. There are many complexities to this, and we will address them one by one.

Simple segmental tandem duplications inside an amplicon can be disambiguated provided the duplication is spanned by long reads. We do not give specific limits on detection in this paper as the read length for Bionano changes depending on instrument, sample prep, other factors. We made numerous changes to the Discussion section (lines 518-526, 561-570) to more thoroughly discuss the reasons for ambiguity and a description of why the “nested duplication” case is sometimes difficult to resolve.

While the reviewer suggests that an additional simulation study could inform our understanding of complex SV nesting behavior, we argue that there are too many variables involved in resolving such nested structural variants to create a meaningful or accurate simulation which simulates such structures from the ground up - especially in the OM case. At least, the following four interacting variables are involved: 1) the size of the nested structural variant, 2) the number of times (n) it is duplicated inside the fCNA, 3) the fraction of reads or maps which span the n copies of the nested variant, anchoring it on either side, and 4) the properties of variation in molecule length, molecule error and assembly error. A very basic simulation would reasonably anchor three of these variables while measuring the effect of varying the other. We argue that the results of this may not be very meaningful as it would only capture the nature of this phenomenon for very restricted set of conditions, without interaction.

Figure R1. Nested duplications can theoretically be resolved with long reads when both errors in graph and CN estimate exist.

Due to those complex interactions between molecule length, assembly ability, repeat number and amplicon size, we deemed a simulation strategy attempting to account for all these variables and their interactions to be combinatorially infeasible. We instead leveraged existing data and used a data-driven approach to fCNA reconstruction which used the state of the art for putative fCNA structure in order to inform the accuracy of our method. We discuss our simulation strategy and new results in more depth in our response to point #3 by this reviewer.

We note that such nested duplications are in fact captured in our current simulation strategy based on suggested amplicon structures derived from NGS data on cancer cell lines (response to point 3c). We report in the simulations that AR resolved 75% (45/60) of these duplications. We reiterate that AR also successfully resolved a segmental tandem duplication of 430 kbp in T47D. The ability to do this however is constrained by 1) length of read/map, 2) length of duplication. Effective resolution occurs when length of read \geq length of duplication (Figure R1).

It is also important to describe the modes by which such a “nested duplication” pattern arises in the context of resolving the biological mechanism of its genesis. We believe aggregated/agglomerated ecDNA (ecDNA which has joined in tandem, as reported in Turner et al., *Nature* 2017) to have the “nested duplication” property (Figure

2b). We cannot reliably distinguish this event from non-agglomerated ecDNA with AR alone. However, it is somewhat immaterial as the reconstruction of a cyclic ecDNA is enough to inform the biological mechanism - whether it is currently agglomerated or not. We address this in more detail in the response to point #2 by the reviewer.

2. “Related to above, a key (biological) question in resolving amplicon structure is whether these structures are circular / extrachromosomal vs chromosomal / integrated / HSRs. A “cycle” on the NGS breakpoint graph can be equivalently interpreted as a tandem segmental dup or a eccDNA. i.e. these patterns will look identical on NGS WGS. Indeed, in practice an (arbitrary, heuristic) copy number threshold is used to distinguish between (lower copy) tandem dups and (very high copy) DMs. OM can potentially distinguish between these two possibilities since the OM maps are on the scale that they may be able to unambiguously place a (reasonably small) amplicon on a chromosome. Do the authors claim that their method is able to make this eccDNA vs integrated seg dup distinction reliably? How confidently can the authors make this assessment? Ideally this would be shown with simulations where the simulated ground truth is eccDNA vs integrated. Such a simulation, if working properly would fail with NGS WGS short reads and would (for a certain event scales) succeed in resolving amplicons with OM.”

Response: The reviewer asks an important question that is central to the field itself. The first is whether we can distinguish ecDNA from structures that were extrachromosomal but aggregated and reintegrated chromosomally. Second, is whether we can distinguish chains of tandem duplications from ecDNA amplification. The first point overlaps a question raised by Reviewer #2 as to where these structures integrate – which we addressed with additional computational analysis in the first response to reviewer #2 and present in Figure R6. The results of this analysis were incorporated into the revised manuscript (Results: “Integration points of focal amplifications”).

To validate our claim that circular genomic structures are indicative of ecDNA, we provide some data from concurrent work (Kim, H. et al. Frequent extrachromosomal oncogene amplification drives aggressive tumors” *bioRxiv* 2019), in which we performed FISH analysis on 81 samples from 54 different cell lines, including 19 glioma derived models, three medulloblastoma models and 32 non-brain tumor models, and described a classification scheme based solely on NGS reads that predicts ecDNA with 83% sensitivity (29 of 35 FISH validated samples were classified as ecDNA positive, Figure R2A). Moreover, 41 of 47 non-circular structures did not show ecDNA in FISH.

We have also tested predictions on primary whole-genome sequencing (WGS) dataset of 15 neuroblastoma tumors, to which the Henssen group also applied Circle-Seq to detect ecDNA (Koche et al, *Nature Genetics* 2019, PMID 31844324). Circle-Seq is a sequencing library enrichment approach optimized for circular DNA detection (Møller et al. *PNAS*, 2015, PMID 26038577). We observed a very high concordance between WGS and Circle-Seq approaches in distinguishing circular from non-circular DNA amplicons (Figure R2B). Specifically, 4 of 65 WGS-detected amplicons were classified as circular by both methods and one of five Circle-Seq derived regions was classified as non-circular. These results translate to a 100% specificity and 80% sensitivity. All 60 amplicons classified as non-circular were not detected by Circle-Seq, implying 100% specificity.

Figure R2. **A.** FISH based counts of extrachromosomal signals per cell, for 81 amplicons classified using AmpliconArchitect. Using 0.5 ecDNA per cell as a cutoff, the sensitivity and predictive positive value for detection of circular DNA from whole-genome sequencing data is respectively 83% and 85%. No-fSCNA – No focal somatic copy number amplification detected. **B.** Summary of whole-genome sequencing versus Circle-seq derived amplicons from 15 neuroblastoma tumors.

Figure R3 Short read strategies can distinguish tandem and non-tandem nested duplications.

breakpoint varies across different samples while amplifying the same gene. This apparent contradiction is resolved if high copy numbers with conserved breakpoints arise due to ecDNA formation. We have added a note explaining this in the discussion section (lines 561-570) which is reproduced in the following paragraph.

“One traditionally difficult fCNA case to reconstruct involves the nested duplication of genomic segments inside an amplicon. Unless a significant fraction of reads or genomic maps have a length greater than the duplicated element, the duplication status may not always be accurately resolved, leading to ambiguity in the possible set of reconstructions. The use of cyclic structures as a signature for ecDNA is based on the reuse of breakpoints. Multiple tandem duplications can also give rise to a cyclic breakpoint graph structure. However, for that to happen, the same breakpoint would need to be reused repeatedly, and evidence points against that possibility. Instead, ecDNA provide a simpler and arguably correct interpretation of cyclic structures, and that has been validated using cytogenetics and comparison to Circle-seq experiments.”

Based on the results of Kim et al., *bioRxiv*, 2020, we can say that even short read amplicon structures provide good evidence of separation between intra-chromosomal and extra-chromosomal amplification (Figure R3). This is, however, a question we do not re-address in this paper. We present AmpliconReconstructor as a tool not specifically for distinguishing between extrachromosomal and intra-chromosomal amplifications, but rather to strengthen the findings of circularity, and to provide better resolved amplicon structure, including improved prediction of mechanisms such as breakage fusion bridge and translocations, and sites of possible integration of ecDNA that have re-integrated into non-native locations.

This leaves the question of structures that were ecDNA but have reintegrated. Indeed, we classify these as ecDNA as our cytogenetic evidence suggests that cells often carry ecDNA that extrachromosomal as well as structures that have reintegrated. As part of the revision for this paper, we have used optical map data to identify sites of integration (Figure R7 below).

3. a) “The simulation seems a bit simple and biased towards the authors' existing tools. Firstly the simulation models molecules rather than whole chromosomes, and not clear whether it is diploid or matches the ploidy of real cancer genomes”.

With respect to the second question, it is important to describe the modes by which a tandem duplication pattern arises in fCNA (Figure R3). Extreme copy number amplification with conserved breakpoints is an indicator of an ecDNA-derived mechanism (See Figure R3, (i)). While one could also explain very large amplifications using tandem duplications, that would require for the same breakpoint to be used again and again (Figure R3, (iii)). However, in other studies (Kim et al., *bioRxiv*, 2020), we have found that the exact

Response: We believe that we may not have properly conveyed the nature of the simulation process and its complexity. Therefore, to address the reviewer's comment, we have created **Figure R4** (on following page, in text as Supplemental Fig. S3) to help address any confusion. Our simulation design captures many critical sources of OM error and is based on real structural variants detected in cancer cell lines. The simulations created individual unassembled OM molecules derived from a diploid reference genome. These individual molecules are sampled randomly from chromosomes and an error profile is applied, using the tool OMSim. The simulation of thousands of individual molecules introduces more realistic sources of error such as OM assembly error, missing labels, etc. We argue this is a superior strategy than end-to-end generation of pre-assembled contigs, where realistic errors do not arise naturally.

The computational time required to generate these simulations was non-trivial. On our cluster computer with dual Intel Xeon E5-2643 (3.5 GHz) CPUs (24 logical threads), 128 Gb RAM, the assembly process of the background reference took four days alone. The total amount of simulated data we generated for all samples in this simulation study was over **two terabytes**. One terabyte for the 85 original simulations + 20 *de novo* simulations discussed in point 3c at approximately 10 Gb per simulation, and another terabyte of data for the 123 heterogeneity simulation examples. In total, the process for generating, assembling, and running AR on these datasets required > three weeks of runtime on three parallel nodes and used the vast majority of our lab's computational resources during that time.

The reviewer raises a very valid question as to whether this is biased in favor of our previous tools. Furthermore, the effect of changing the background reference genome to more closely mirror a real cancer genome is a point well taken. We respond to that in the answer to point 3c, where we introduce a tumor reference into *de novo* simulations of circular ecDNA.

b) "Secondly, the simulated chromosomes are currently based on paths outputted from AA breakpoint graph, which are likely short and fragmented since they derived from NGS WGS? These graphs also likely do not have connectivity in repeat regions since the AA graphs would miss these SV's in the short read NGS data."

Response: While it is true that our simulated amplicons are based on AA results, the role of AA in this simulation was to create an underlying set of realistic structures from which OM molecules were sampled and simulated. Importantly the molecules simulated from these structures were added into a set of molecules generated from whole genome simulation. Thus, the simulations capture the complexity of the whole genome and amplicons combined. The average length of simulated optical map molecules in our study was 150 kbp.

The paths in our simulation have a median length of 1.1 Mbp and contain a mean of 17.5 genomic segments, which we believe are similar to the size and complexity of real focal amplifications in cancer, and importantly contain many repeat elements such as LINEs and SINEs (Deshpande et al. *Nature Communications* 2019). AA uses a database to filter low-complexity regions from SV analysis which is derived from the database used by another state-of-the-art SV caller, Lumpy. The issue of missed SVs in low complexity regions is a standing problem in the SV field. We note that AR can use the Bionano data to enable the discovery of SVs missed by AA in such regions.

We are not aware of a diverse set of possible reconstructions of focal amplifications which are **not** generated by AA – and if we had been able to identify another tool which output such data it would possibly be biased in favor of that tool's results. However, to address the reviewer's concerns, we designed a new set of "ground-up" simulations to address two points. This simulation strategy is described in the response to point 3c.

Figure R4. Flow diagram for AR simulation strategy. Either hg19 or a simulated tumor reference based on hg19, using SCNVSim was used. Simulated amplicons were either derived from prior AA results or created *de novo* by simulated amplicons using ecSimulator.

c) “Finally, it's unclear whether the AA-derived simulated rearrangements would have the sort of nested (tandem or inverted) duplication structures that would make these kinds of reconstructions truly challenging in practice. It would be more convincing to use a simulation that rearranges a fully diploid genome end to end and explicitly models a BFB and complex double minute. This would also help determine when / how well AR can resolve essential features of amplicons (ie chromosomal vs ecc).”

Response: This is a good point raised by the reviewer, and in addition to answering the reviewer's questions we include a description of new simulations performed to address this point.

In response to the point as to whether it is unclear if these structures contain internal duplications, we analyzed the duplications present in our simulation set. Across the 85 original examples plus the 20 *de novo* simulated circular ecDNAs discussed in the following paragraphs, 60 duplications of chains of breakpoint graph segments were present in the simulated amplicons. The average size of duplications present in these examples was 281 kbp (minimum 224 base-pairs, max 849 kbp). AR resolved the duplication status in 75% of cases (45/60). We have clarified this point on lines 189-191, reproduced below:

“Large duplications inside a rearranged amplicon represent a challenging case to reconstruct. We identified 60 duplications of one or more graph segments (mean length 281 kbp) in the simulated amplicons, and we report that AR resolved 75% (45) of these duplications.”

The initial simulations we performed **did** contain circular paths (double minute structures). In total, 44% (37/85) simulated paths were circular. These were of course based on putative circular ecDNA structures as detected by AA with NGS data from real cancer cell lines. We have clarified this on lines 152-155.

As AR does not perform an automated analysis to assign a biological mechanism (ecc vs chromosomal), as described in the Discussion section, that part of the process requires manual interpretation. Ultimately, the simulation study measures the accuracy of AR's reconstructions in the aggregate, across amplicons generated by many mechanisms.

To address the point related to an end-to-end model which explicitly models complex DMs, we designed and implemented the following strategy.

- 1) In the original simulations, the simulated molecules from the background reference (non-amplicon) were generated with hg19 and assembled. In our new set we instead used SCNVSIM, a tumor genome simulator capable of generating SNVs, CNV/ploidy changes and structural variants on a flat reference to create a simulated cancer reference genome with hg19 as input.
- 2) The original simulations used putative AA reconstructions from the 2019 paper, which were used as input for simulating OM data. To provide an unbiased alternative to this, we created a ground-up simulation utility, called ecSimulator (<https://github.com/jluebeck/ecSimulator>), which generates random extrachromosomal DNA structures given some user-specified parameters.

EcSimulator is a new utility we are developing that selects random intervals from the reference genome and assigns breakpoints along those intervals. It then conducts SV operations on those breakpoints, including duplications, deletions, inversions and translocations.

Importantly, we note that AR had no statistically significant difference in performance (measured by F1 scores using the “Length (bp)” metric between the original simulations and the *de novo* simulations. This suggests that

our simulations are likely not biased towards AA. As we state in the updated results section: *“In this final simulation study we created 20 new amplicons which were subject to our simulation pipeline. The simulated amplicons had a median size 2.0 Mbp, and mean segments 9.3. We replaced the hg19 reference used to generate the background molecules with a simulated tumor genome generated with SCNVSIM. AR’s performance on these cases achieved a mean F1 score of 0.860 (0.731 when assembly failures included, Supplemental Table 2). The distributions of F1 scores for the 20 de novo cases and the 85 AA-derived cases as measured with the “length (bp)” metric were not statistically different between the 85 AA-derived amplicons and the 20 de novo simulated amplicons (two-tailed Mann-Whitney U-test, p-value = 0.1996, test statistic = 631.0).”*

BFB has classically been detected through signatures of BFB (foldback reads, FISH, etc), not through reconstruction of the BFB itself. As a result, the field lacks wide-spread and detailed knowledge of real BFB structures, from which to perform a data-driven simulation and thus we do not explicitly model a BFB structure in our simulations. In this paper we are not attempting to make an automated prediction of the fCNA’s biological mechanism but rather to resolve the fine structure as accurately as possible. The resolved fine structure can enable a user to interpret the underlying biological mechanism, as we did with BFB where we were aided by the established theoretical model for BFB formation produced by Zakov et al.

4. a) “Can the authors draw a better conceptual comparison / contrast of AR vs AA, including comparing results? I get it that AR incorporates OM data, but don’t the tools otherwise have the same goal? It seems that the authors only utilize AA for breakpoint graph inference, but if I recall AA also generates paths. How do the AR vs AA paths compare? Similarly, AR integrates the NGS and OM data and presumably also should create a modified (improved) breakpoint graph, which better account for copy number changes (see below). How do the AR vs AA breakpoint graphs compare? These comparisons would help provide intuition as to the value added by OM.”

Response: This is a great question. AA indeed produces a cycle-decomposition of the breakpoint graph – which represent possible reconstructions based purely on NGS data. We have conducted new analysis to answer this question and described the results presented below (Figure R5) which compare AA and AR under a subheading in the Results section, “AR provides a reconstruction improvement over AA.” We have reproduced the added section below:

“AA can attempt to identify some heaviest weight paths and cycles in the breakpoint graph only using NGS data. We demonstrate that for complex amplicons, AR provides an improvement to the fraction of the amplified genomic segments which are explained by the output structure compared to the heaviest path or cycle generated by AA (Supplemental Fig. S15a). As OM data may suggest additional amplicon junctions not observed in NGS, we observed that the resulting set of amplified segment junctions observed in the AR output was equal to (GBM39, T47D) or larger than (CAKI-2, H460, HCC827, HK301 and K562) the number of junctions suggested in the AA breakpoint graph alone (Supplemental Fig. S15b).”

One primary difference however is that AR does not attempt to construct a breakpoint graph. In fact, AR does not generate a new breakpoint graph as output. It uses the AA graph to identify higher-quality scaffolds and these scaffolds may infer junctions not observed in the AA breakpoint graph. AR can be thought of as method to simplify the breakpoint graph and identify chains of amplicon segments.

Figure R5. **A.** Proportion of amplified segments in the AA-generated breakpoint graph which are explained by AA and AR heaviest reconstructions. **B.** The number of breakpoint edge junctions inferred by AA and by AA + AR (union).

To show how the paths suggested by AA (with NGS alone) and the paths suggested by AR compared, we created a supplemental figure shown in Figure R5 (Supplemental Fig. S15a,b). This figure shows the proportion of the breakpoint graph explained by AA and AR (**A**) and the contribution of AR to adding junctions to those identified by AA (**B**). Some samples such as GBM39, H460 and HK301 show very similar performance. It is important to keep in mind that samples were selected non-randomly. Those were less complex samples included to help make the development process of AR more feasible. The more complex samples (K562 and HCC827) show better performance with AR.

b) “e.g. Figure 2A shows a breakpoint graph with many apparent copy number changes that are not associated with a structural rearrangement. This is presumably the NGS WGS graph inferred by AA, but AR should improve this graph? How well does it improve the graph? Can such an AR-improved graph account for all the copy number changes with a rearrangement or are there still links missing?”

Response: As Figure 2A indicates a FISH image, we will assume the reviewer is referring to Figure 3A, which does indeed show the copy number changes without structural rearrangement. We created Figure R5 to address the first two

questions asked by the reviewer, and this is also included in the subsection “AR provides a reconstruction improvement over AA”.

In the case of K562, AR did suggest additional junctions between amplified segments, not contained in the breakpoint graph. With OM alone such junctions are necessarily coarse-grained due to the mapping resolution of OM. AR suggested 25 additional junctions beyond the 30 suggested by AA alone (55 total). There may still be links missing from the breakpoint graph which AR did not detect, but that is not possible to quantify in this case. We have added a line to clarify this in the Discussion (lines 152-155).

With regards to the last question, while AR reconstructs an amplicon containing all amplified segments, it does not explain the variable copy numbers between chr9, chr13 and chr22 as the reviewer pointed out. Presumably there is some level of heterogeneity present. AR is designed to reconstruct the dominant focal amplification and detecting lower abundance fCNA is outside the scope of this paper. Regardless, we agree the CN variance not explained by AR should be better clarified and this is addressed in the revised version of the Discussion section on lines 327-330 and 543-552, including “... *Copy number variance in this amplicon not explained by AR may be due to structural heterogeneity across the many copies of the amplicon...*”

5. “The paths that are inferred by AR presumably imply a copy number, and then each path has a multiplicity. When integrated together, does this copy number match the total copy number of the region? It seems like the method only uses the total copy number as an upper bound? If so, then how do the authors explain the remaining copies and breaks in the region?”

Response: AR examines the ratio of copy numbers (Methods: “Finding reconstructions in the linked scaffold graph”). Thus, if a segment is duplicated inside an amplicon and thus has 2x CN above the rest of the focal amplification, it can have 2x as many copies in the resulting reconstruction. The method used by AR is not constrained solely by the maximum copy number, but the ratio of copy numbers in the breakpoint graph. We allow for some error to occur in the CN reported in the breakpoint graph. The number of copies implied by the ratio may exceed the number of breakpoint graph copies by 1, to account for any underestimation in the NGS-derived copy number. However, as the reviewer points out, there may be excess copy numbers present in the graph which are not explained by the amplicon. Unexplained CN may be due to heterogeneity of the amplicon in the genome, missing breakpoints, missing segments, issues with OM assembly or some combination of all these factors. We have clarified this in the discussion section on lines 544-546, 554-560, including:

“The collection of paths reconstructed by AR represent possible reconstructions of the fCNA, and the collection of paths may contain multiple similar explanations for the fCNA architecture. This may be in part due to genomic heterogeneity, limitations of the optical map assembly process, or errors in linking scaffolds across overlapping graph segments. Furthermore, technological limitations related to the quality of OM assembly may affect the ability to reconstruct high-fidelity amplicons.”

6. a) “How did the authors arrive at the reconstruction of the time sequence of events for the BFB amplicon. Was this done manually or is it a part of AR? If the latter can the authors reveal the method? Are there alternative reconstructions? Otherwise, it would be clearer to emphasize that this part of the analysis was manual and not an output of AR.”

Response: We agree that these are important questions. The reconstruction of the time sequence of events for the BFB amplicon was performed manually, using the theoretical underpinnings in Zakov et al. We found no other alternative reconstructions of the time sequence which were consistent with the theoretical model for BFB formation. We have clarified this point in the manuscript: *“When the AR scaffolds were combined with the copy number data present in the breakpoint graph, we were able to manually identify a single BFB structure, that was consistent with the theoretical model of BFB formation.”* (lines 395-397).

b) “where does the BFB “end”? Does it acquire a telomere of another chromosome or is a new telomere synthesized at one of its ends?”

Response: We believe that the BFB repeat unit we reconstructed is repeated 10 times between the of the BFB ends (centromere to telomere of chr7p) based on the copy number of the segments in the breakpoint graph and the reconstructed repeat unit. As discussed in Maciejowski and de Lange, *Nat. Rev. Mol. Cell Biol.*, 2017, BFBs often reacquire a telomere. However, as noted in Carroll et al., *Mol. Cell Biol.*, 1988, in some cases a BFB will end by making a DM (ecDNA). We did not detect any DM-like reconstructions in this case and did not observe any EGFR-bearing DMs in the HCC827 FISH results. We have added this negative finding for BFB-derived double-minute structures in the Results section: *“While some BFBs may result in “double-minute” amplicons, AR suggested and FISH analysis confirmed that the HCC827 BFB does not contain a circular extrachromosomal version of the BFB cycle.”* (lines 391-393). We therefore conclude that the HCC827 BFB

stabilized with the acquisition of a telomere. AR is not designed to do telomere detection, and we feel attempting to detect the telomere capping the BFB is an analysis that is outside the scope of the paper.

Reviewer #2 (Remarks to the Author):

“Their approach and algorithm appear to be adept at reconstructing fCNA architecture, even if currently limited to samples with both NGS and OM data. The manuscript is well written, and highlights both the advantages and caveats to their approach, which is appreciated.”

We thank the reviewer for this comment.

1. “Given the great interest in the field in computationally distinguishing between extrachromosomal and intrachromosomal amplifications, can the data/approaches here offer any insight into whether HSRs might be genetically joining the linear chromosomes to which they appear to be physically attached as observed in FISH? As the authors point out, the differences in topology between ecDNA and HSR-like fCNAs don’t lend themselves to differences in the breakpoint graphs, but these breakpoint graphs are in fact derived from the focal amplifications mapped in AA, correct? If HSRs are at all directly fused to the linear genome (and they may not be), are there any ‘edge’ cases demonstrating something resembling an ‘exit’ from the fCNA to the associated linear chromosome in NGS or OM data?”

Response: This is a very good question and represents an area of the field which is understudied. Low frequency breakpoint edges, such as the ones indicating integration points may not appear in the NGS breakpoint graph and they may not get incorporated into assembled OM contigs. We used Bionano Solve molecule alignment methods for OM molecule (unassembled) to search for evidence of such integration points in NGS and OM molecules.

In CAKI2, H460 and K562 we identified four estimated integration points with 10 or more molecules of support in the OM data. To identify these points, we sorted aligned molecules and clustered alignment endpoint pairs. The values below represent the center of the cluster – and not a refined breakpoint.

Unfortunately, HK301, the sample where FISH suggested an integration of a circular ecDNA, did not have high enough OM coverage (> 100, Supplemental Table 1) to perform such an analysis. We have described this analysis in the manuscript under a subheading in the Results section, “Integration points of focal amplifications”, reproduced below:

“Using alignments of unassembled optical maps, generated by the Bionano RefAligner molecule alignment pipeline, we gathered molecules with split alignments joining a partial alignment inside the amplicon region with a partial alignment outside the amplicon region. Low frequency breakpoint edges, such as the ones indicating integration points may not appear in the NGS breakpoint graph and they may not get incorporated into assembled OM contigs. For H460, K562, CAKI-2, HCC827 and T47D, OM coverage was deep (> 100, Supplemental Table 1) allowing us to cluster the split alignments into OM-derived breakpoint clusters suggestive of low-frequency integration points. Requiring that each breakpoint cluster have 10 or more molecules suggesting the same split alignments (within 25 kbp on either side), we were able to identify course-grained integration point candidates. The breakpoint cluster centers are reported in Supplemental Table 3.

Figure R6. Genomic integration points detected in the samples analyzed by AR.

Visualized with MapOptics, H460 showed a single integration point between amplicon region chr8:129410000 and non-amplicon region chr12:7660000 (Supplemental Fig. S14a). K562 showed two integration points. The first joined amplicon region chr13:81120000 and non-amplicon region chr1:142890000 (Supplemental Fig. S14b). The second joined amplicon region chr13: 93260000 and non-amplicon region chr1: 142890000 (Supplemental Fig. S14c). The proximity of these two integration points suggests a left and right boundary for the integration of the K562 BCR-ABL1 amplicon. CAKI-2 showed one integration point joining amplicon region chr12:88300000 and non-amplicon region chr6:168380000 (Supplemental Fig. S14d).

HCC827 and T47D did not show any such integration points with 10+ molecules of support, which is consistent with the finding that these were chromosomally derived focal amplifications (BFB and segmental tandem duplication respectively) residing in their native locations.”

The breakpoint cluster centers from Supplemental Table 3 are provided below.

Sample	chrA	PositionA	chrB	PositionB	support (mols)
CAKI2	chr12	88302814.5	chr6	168382883.2	25
H460	chr12	7660219.278	chr8	129414270.1	18
K562	chr1	142891183.2	chr13	81123336.19	326
K562	chr1	142888366.9	chr13	93260823.16	50

2. “As the authors point out, a previous effort integrated NGS (WGS and HiC) and OM data to study structural variants (Dixon et al, Nature Genetics 2018). The Dixon et al analysis of data from the renal cancer cell line CAKI-2 yielded a chr2-chr3 fusion (Fig 1) as well as a chr6-chr8 fusion (Fig 2), whereas AR results on CAKI-2 in this paper yielded a segment of chr3 joined to chr12 (Supplemental Fig. S11c,d). It’s clear that different approaches will always have certain advantages and disadvantages, but can the authors explain the discrepancies here? Also, given the increasing abundance of HiC maps, might they be used in future iterations to aid the AA if not the AR steps in the current approach?”

Response: We agree that a comparison with the Dixon et al. paper is very prudent. We had initially felt that such a comparison was unfair as our methods enable reconstruction of chained breakpoints only in focally amplified regions, while they searched for all genomic breakpoints.

To address this point in a fair manner, we examined the breakpoints reported by Dixon et al. which overlapped the regions we studied. While they reported breakpoints using a number of different technologies separately, we used their high-confidence integrated breakpoints (Dixon et al., 2018 Supplemental Table 6) in our comparison. We have included the results of this analysis in a subheading inside the Results section, “AR provides a reconstruction improvement over AA.”, reproduced below:

“A study by Dixon et al. used an integrative approach to detect structural variation and associated breakpoints using a combination of NGS, OM data and other sequencing modalities. We identified four cell lines shared between this study for which Dixon et al. provided the breakpoints identified by their integrative approach. We observed that in the regions analyzed by AR, more breakpoints were detected with AR than with the integrative approach, though there were some breakpoints indicated by Dixon et al. which were not observed in AR (Supplemental Fig. S15c). In those cases, the majority of breakpoints not observed by AR joined amplicon

regions to regions outside the original amplicon (CAK1-2: 2 of 3 non-AR breakpoints, H460: 1 of 1 non-AR breakpoints, K562: 11 of 16 non-AR breakpoints). In H460, the one breakpoint not observed by AR was the integration point we later detected, suggesting that these are typically lower frequency breakpoints perhaps related to integration or heterogeneity.”

We identified four cell lines shared between the two studies with breakpoints reported in this table; CAK12, H460, K562 and T47D. As they performed their analysis entirely with respect to hg38 and we used hg19, we first lifted over all their breakpoints on the same chromosomes as our amplicons to hg19. We then matched breakpoints from Dixon et al with breakpoints in the AR reconstructions and counted the amount of overlap.

Figure R7. Overlap between detected breakpoints by Dixon et al. and AR within the focally amplified regions analyzed by AR.

In all four cases, AR identified more breakpoints in these regions than Dixon et al. (Figure R7). This may be because AR provides a level of validation to structural variants suggested in NGS in addition to the large ones OM detects on its own. In this regard the NGS and OM data have a synergistic ability to detect breakpoints when combined in our study. In comparison, the Dixon study requires multiple technologies to independently identify the same breakpoint, and thus the presence of one breakpoint in one technology does not help to condition the existence of the same breakpoint in another technology. These findings are reported in the manuscript in the Results section, in the last paragraph under the subheading “AR provides a reconstruction improvement over AA.” Figure R7 has been added as Supplemental Figure S15c.

We do note that the existence of breakpoints identified by Dixon et al. which are not present in our AR reconstruction despite overlapping the amplicon region(s). This may be because AA, which is responsible for building the breakpoint graph assess the existence of breakpoints in a manner which considers the copy number of the region. Thus, some breakpoints that occur in low frequency may not appear in the AR reconstruction.

Hi-C is a promising way to detect large structural variants perhaps missed by traditional NGS. It would provide a very valuable addition to AA and we will consider adding that ability in the future. As for the possible contribution of Hi-C data to AR, we feel AR would best be served by incorporating new technologies that can chain multiple breakpoints together with single reads, simplifying a breakpoint graph. Thus, we feel Hi-C would be worth adding to AA, but not necessarily to AR. We added a note mentioning this point in the discussion section “Other sequencing modalities involving NGS with modified sample preparation, such techniques based on Hi-C, have shown the ability to reveal additional genomic breakpoints without the need for an additional sequencing instrument. While constructing a breakpoint graph is not a part of the AR algorithm, we acknowledge that such techniques would be valuable to adapt for breakpoint graph generation.” (lines 580-585).

3. “In the authors’ 2019 paper (Deshpande et al), they use Amplicon Architect in an innovative way to find human-viral fusion segments in cancer genomes and explore their functions and potential origins. With the OM

strategy, how might incorporation of these additional (non-human) reference sequences work with the AR approach?"

Response: This was not something we had initially considered but we believe AR would certainly be useful in validating the integration of certain oncoviruses into the cancer genome. Viral genomes are generally too short to be reliably labeled with OM fluorescence (typically 10 kb or less). Thus, OM data can serve as a validation of integration, but on its own is very unlikely to be useful - unless the OM protocol is modified to provide a separate fluorescence on a custom-designed label for a viral sequence. We decided to address this question by performing an additional simulation.

Figure R8. AR reconstruction of a simulated ecDNA containing HPV16.

To demonstrate that AR can validate a viral integration suggested by NGS, we performed a simulation (following the strategy in Figure R4) where we simulated a circular ecDNA with human papillomavirus-16 (HPV) viral integration (sequence length: 7906 bp), similar to such structures reported by Nguyen et al., *Nucleic Acids Research*, 2018, and Deshpande et. al, on top of a simulated tumor reference genome background. AR was able to identify the viral integration successfully (Figure R8) despite the viral sequence having no labels, using the adjacencies suggested in the simulated breakpoint graph. We note that experimental strategies which separately label the HPV16

genome during OM sample prep may also provide a high throughput method for integration location identification. We have added this new result to the Results section, in the last paragraph of the "AR reconstructs ecDNA in multiple forms" subheading, reproduced below:

"We had previously documented a circular amplicon containing an integrated human papillomavirus-16 (HPV16) genome, and we hypothesized that AR could be used to help resolve the location of viral insertion in the host genome. We simulated a 1 Mbp circular amplicon with the 7.9 kbp HPV16 genome randomly inserted. AR was able to reconstruct the structure of the circular ecDNA as well as the integration point of the HPV16 sequence (Supplemental Fig. S7), despite the viral genome having no OM labels, suggesting that AR would serve as useful method for validating the existence of genomic oncovirus integrations suggested by NGS data."

4. "The authors state that future iterations of AR may involve input from other long-read sequencing technologies. Perhaps outside the scope, but given that they've worked with PacBio data in the past (Deshpande et al 2019), can they comment on read lengths/accuracies that would make such efforts worthwhile for increasingly used platforms like PacBio and Nanopore?"

We suggest that read lengths in excess of 150 kbp will span two or more breakpoints (with enough overhang for sufficient anchoring) for the majority of fCNAs, given that the median length of genomic segments in our study was 100 kbp. As the resolution of OM data is so coarse, we consider nanopore data to be less noisy than OM data and thus we felt it goes without saying that nanopore is accurate enough for this problem. Given that modified nanopore protocols can routinely generate a substantial fraction of reads in excess of 150 kbp, we suggested that nanopore would be valuable for fCNA resolution and we have added that to the Discussion section at lines 575-578. We intend to adapt AR for nanopore in the future. We feel however, that PacBio reads are still too short for reliable reconstruction of fCNA (very few reads > 100 kbp - <https://www.pacb.com/products-and-services/sequel-system/latest-system-release/>). The fCNA reconstructed in Deshpande et al., using a combination of NGS and PacBio was only 100 kbp.

“With modified protocols, nanopore reads may routinely surpass 150 kbp in length, which is sufficient to frequently chain multiple breakpoints in fCNA. When paired with NGS data we hypothesize that one could modify AR and achieve similar results to NGS and OM data. We plan to address this point in future methods development.”

5. “Given that all of the data in this manuscript has been generated from cell lines, can the authors comment on challenges they expect to encounter when attempting to reconstruct amplicons in primary patient data with this approach?”

This is an interesting question to consider. We have addressed this in the Discussion section of the manuscript on lines 536-541, reproduced below:

“While we analyzed data from cancer cell lines, sequencing data collected from patients may introduce more sources of complex genomic structural heterogeneity. Using AmpliconArchitect, we have previously analyzed > 3000 whole genome sequences from primary tumors and achieved similar success in resolving ecDNA status as with cell line data, and similar levels of heterogeneity as measured by breakpoints per amplicon.”

Reviewer #3 (Remarks to the Author):

“This is a well-written paper that describes an exciting new piece of software AR. AR can combine sequencing (WGS) and optical mapping (OM) data to accurately reconstruct the rearrangement events that commonly occur in various tumor lineages. Specific examples of various rearrangement and copy number amplification mechanisms were inferred and illustrated nicely. I believe that AR would be a useful software for researchers aiming for accuracy and resolution of such events. There are still some issues though as I went through this paper.”

We thank the reviewer for this comment and for the reviewer’s exceptionally close reading of the manuscript.

1. “I think one of the immediate questions a reader might have – what is the added benefit of OM on top of WGS? The authors need to make it explicit during the Discussion. There have been many separate studies looking at WGS alone, or OM alone, what are they missing? AR makes heavy use of the breakpoint graph from WGS, does it somehow limit AR from the discovery of additional events that are difficult to identify using WGS (e.g. lack of coverage for certain regions, variation in mappability across the genome). There are some brief

Figure R5. **A.** Proportion of amplified segments in the AA-generated breakpoint graph which are explained by AA and AR heaviest reconstructions. **B.** The number of breakpoint edge junctions inferred by AA and by AA + AR (union).

hints of this mentioned in the Results but not summarized systematically which I think would be highly valuable. Similarly, what are the added benefits of WGS+OM on top of OM alone?"

Response: This is an important discussion point not clarified thoroughly enough in our original manuscript. We answered a very similar response to a similar question by Reviewer 1 (point #4 a,b). In comparing AA's reconstructive abilities based on NGS alone with AR (AA output + OM) we note a marked improvement in reconstructive ability (Figure R5). One way to view AR is that AR provides a level of validation to structural variants suggested by NGS.

The manuscript describes that the median molecule ("map") length used in assembly across all samples used in this study is 244 kbp (molecule N50 340 kbp), while the median segment length in breakpoint graphs used in this study is 100 kbp, highlighting that OM data can span multiple junctions in the fine-resolved breakpoint graphs derived from focal amplifications (Supplemental Table 1). By virtue of OM not being "sequence-based", the integrated NGS data and OM data provide an orthogonal pairing of short- and long-range information about genomic structural variation. In Chaisson et al. *Nature Communications* 2019, the authors demonstrate the increased ability of OM to detect extremely

large structural variants over NGS. While NGS provides finely mapped breakpoints, if the exact breakpoint has poor mappability, the breakpoint may go undetected.

To expand on this point, the NGS and OM data have an independent ability to detect breakpoints when combined in our study. At the same time, we use the OM junctions to condition the existence of the suggested NGS breakpoints. In that regard there is a synergistic relationship between NGS and OM when used in AR. In comparison, the Dixon et al. *Nature Genetics*, 2018 study requires multiple technologies to independently identify the same breakpoint, and thus the presence of one breakpoint in one technology does not help to condition the existence of the same breakpoint in another technology.

We show the added benefit of WGS+OM over OM alone is that, without NGS data, imputation of short segments into OM scaffolds would be impossible. We show that improvement as a simulation result (Fig. 1i and Supplemental Fig. S4). Most tools (either for NGS or OM) look for individual SVs and do not try and capture amplicons structures. The reconstruction of BFB and complex ecDNA would not have been possible without the combination of both modalities. We have edited the Discussion section to address these points (lines 508-511):

Figure R9. Precision (top) and recall (bottom) for AR + SegAligner. CN 20 in solid blue. Red boxes indicate region where precision < 0.6, recall < 0.6.

“OM tends to detect larger SVs than NGS alone and is less affected by mapping issues on low complexity breakpoints. We have demonstrated that NGS data, when incorporated with OM can be used to resolve fine-mapped breakpoints suggested by OM.”

2. “Some in-depth analyses may be needed to look at the results from simulation studies to understand the common error modes. What are the characteristics of the false positive or false negative, or poor LCS predictions from AR? While this reviewer has little doubts that AR would be useful – it is important to inform the readers when the software fails and how they fail. Would supplying more input data (both the WGS and OM) help in those cases? The mixture experiment on heterogeneity is very nice. Can we learn what is the limit of capability of detection from the mixture experiment? For example, in terms of the minimum level of abundance or the number of clonal subpopulations in the sample.”

Response: We agree that we should present a better understanding of the relatively small number of cases where AR shows poor results on simulated data.

To produce a more complete understanding we manually analyzed the results from the AR + SegAligner simulation. In the AR + SegAligner simulation, both modules were our tools as opposed the cases where RefAligner or OMBlast were used for OM alignment. Specifically, we analyzed the simulation that had path imputation enabled (disabled imputation was only simulated for purposes of showing the improvement imputation yields) for amplicons with CN 20.

Considering all simulated amplicons where precision was < 0.6 (N = 13 out of 85 simulated structures):

- 9 cases showed signatures of assembly failure (i.e. the Bionano Assembler did not reconstruct any contigs covering the region of interest which did not differ from the reference genome).
- 3 cases showed signs that at least some amplicon contigs were assembled but the breakpoint graph was too complex/segmented for AR determine a correct structure. A highly segmented breakpoint graph leads to difficulty identifying “anchor” segments to form the backbone of a reliable scaffold using the OM data.
- 1 case showed that the contigs were correctly assembled and the individual scaffolds generated by AR was correct, however the linking of the different scaffolds on unbroken graph segments was incorrect – leading to an incorrect reconstruction.

Considering all cases where the recall was < 0.6 (N = 14 out of 85 simulated structures):

- 9 cases showed signatures of assembly failure.
- 5 cases showed signs that the breakpoint graph was too complex.

Regarding the question as to whether we can learn what the limit of focal amplification is from further mixture experiments - We are confident that the selected mixtures capture the levels of heterogeneity which are both detectable and still at reasonable copy numbers to be considered as focal amplifications. We believe that there are too many variables involved to design a meaningful simulation that would accurately identify the limit of detection from a mixture experiment. We have added a note in the results to explain some of these findings (lines 198-207), reproduced below:

“To understand the reasons for loss of performance on a small number of simulation cases, we examined the results from the CN 20 simulation where individual reconstructions showed either precision or recall < 0.6. We manually examined the results from the 85 total cases and found that of the 13 amplicons with precision below this threshold, nine cases showed signs of assembly failure, while three had incorrect reconstructions likely on account of graph complexity. The remaining case showed an issue with incorrect scaffold linking. Of the 14 amplicons having recall below the threshold, nine cases showed signs of assembly failure, while five had highly segmented breakpoint graphs making it difficult for AR to identify anchoring alignments around the breakpoints, leading to an incomplete reconstruction.”

3. “In the Introduction, please introduce the key algorithmic challenge and what is the main algorithm and statistical models employed in tackling this challenge. Please mention any prior statistical modeling work on the properties of NGS-read breakpoint graph and optical mapping data. For example, the source of variation on segment size, missing labels, uncertainties in the direction and connection within the breakpoint graph, etc. to provide some theoretical ground for the AR method later in the paper.”

We have modified the Introduction section to draw more attention to the key algorithmic challenge; *“ordering and orienting multiple genomic segments joined by breakpoints into high confidence copy number-aware scaffolds, which are subsequently joined to enable complete reconstructions of complex rearrangements.”* (lines 71-74). We feel that discussion of the main algorithms we developed to address the computational problem should be done at the beginning of the Results section. We have reworded the opening paragraph in the Results section to frame these new algorithmic developments more clearly (lines 120-140).

“We formulated the problem of fCNA reconstruction in multiple parts. First, alignment of genomic segments with optical map contigs. Second, the reconstruction of a genomic scaffold using OM data as a backbone. Third, the identification of the maximal simple paths in a graph where each node is an OM scaffold, for which the path is not a sub-sequence of another maximal simple path. AR separates these computational tasks into four primary modules (Fig. 1a,b)... [description of each module]”

We note that there is only limited statistical modeling of the breakpoint graphs we use as input. We provided some of the properties in the Results section where we profile the breakpoint graphs we simulated (line XXX): *“These included both cyclic (37 paths) and non-cyclic paths (48 paths) with lengths varying from 260 kbp to 2.8 Mbp (median 1.1 Mbp) and the number of graph segments varying from 3 to 47 (mean 17.5 segments; Supplemental Table 2).”*

There is more prior modelling however related to the properties of optical mapping. We added the predominant sources of OM error to the introduction (lines 95-98) and cited two recent papers which discuss mathematical models for Bionano data and errors (Li, M. et al. “Toward a more accurate error model for BioNano optical maps” *Springer Verlag*, 2016 and Chen, P., et al., “Modelling BioNano optical data and simulation study of genome map assembly” *Bioinformatics*, 2018). The introduction of the Bionano Saphyr system has recently

changed many of the error rates downwards for optical mapping, providing longer, more accurately labeled molecules. Given the continuing improvements to throughput and sample prep made by Bionano, we feel that performing a complete quantification of the error rates for all different modes is outside the scope of this paper.

4. “There were a number of issues in the Methods section. Please check the formulas and pseudocode more carefully. While I could conceptually follow most of it, I had some issues going through.”

a) “Check DP recurrence equation, especially the sub for the max (line 578), I don’t think I agreed with the current upper bounds for i and p .”

Response: There were indeed two typos in the upper bounds. The recurrence has been corrected.

b) “Poisson distribution (line 658), should it be a minus or equal sign? The formula also seems wrong for the exponential of e part. Should it be $-E$ instead of $-a$?”

Response: The reviewer is correct. We have corrected this typo.

5. “Algorithm 2: explain what c and k are and whether they are fixed parameters or estimated from the data (and how). What is the rationale behind this scoring function? Specifically, what is the role of each of these terms? Explanations would help.

“Some parametrization needs a little more explanation. Explain why different modes of alignment warrant different levels of P-value threshold, and how are these thresholds determined. Parameterization of Irys and Saphyr is different due to their properties (e.g. Fig. S1d) – Supplementary Table 5 contains at least 5 parameters that are different, as well as the minimum number of labels (line 671). What is the process to tune these parameters per platform, at least on a high level? This will be relevant as Bionano continues to upgrade the chemistry and specs, and for people trying to optimize those parameters for their own data.”

Response: We thank the reviewer for raising these important points. We have added an explanation of the scoring function, its terms, and constants c and k in the Methods section on lines 724-733 and reproduced the explanation below this paragraph. We tested many possible values for c and k during development, through a grid search approach on data with known OM alignments and these gave the best performance based on optical mapping tests we designed. We have also tried much more complex scoring functions for OM alignment. For example an early version of SegAligner we built used a likelihood ratio-based scoring model (as used in Valouev et al., *Journal of Computational Biology*, 2006) with parameters learned from the data using a maximum likelihood approach, yet we ultimately found our simpler heuristic formulation presented here performed the best. OM data errors are still not as well understood as errors in other sequencing modalities and hard to model (compared to NGS or even Nanopore), thus a heuristic worked better for us than a probabilistic scoring model.

“Score is defined in Algorithm 2 and contains four main terms. First, f_n which is defined as the number of potentially unmatched contig labels between i and j scaled by the missing label score, c . Second, e_{ref} is the number of potentially unmatched reference labels between p and q , after accounting for labels which are too close together to be measured distinctly. Third, f_p is the number of potentially unmatched reference labels scaled by the missing label score. Lastly, Δ measures the absolute difference in length between $j - i$ and $q - p$, which is scaled non-linearly (k). Together these penalty terms are combined and subtracted from a base

matching score 2c. Parameters c and k in this model were identified through a coarse grid-search using data where correct OM contig-reference alignments were already known.”

The different modes of alignment use different p-value thresholds as they consider search spaces of different sizes, and thus we control the false discovery more stringently for larger search spaces. The detection of reference segments, which involves alignment against the entire genome has the largest search space and thus gets the smallest p-value threshold. To set the default values, we used results of SegAligner on data where OM alignment was already “known”, i.e. identified through a combination of Bionano RefAligner, OMBlast, and extensive manual inspection. We have described this on lines 824-829.

“The need for different p-value thresholds between the different modes of alignment is based on the different sizes of the search spaces possible in the different modes. Searching for alignments between contig and entire reference is the largest search space to consider and thus it gets to smallest p-value threshold in order to stringently control false discovery. The default p-values for each mode were assigned based on empirical testing of OM data with known alignments.”

Our decision to parameterize the instruments differently was a consequence of the improvement in label resolution, and the changes in labelling chemistry and labelling density of the DLE1 (direct labeling) recognition sequence vs BspQI (nickase) as observed in the NA12878 data released by Bionano. Saphyr data tends to be better resolved, having less directional uncertainty and a smaller rate of label collapse (as reflected by parameters t,w, and η in Supplemental Table 5). Again, we tuned these parameters by running SegAligner on data with known alignments and using coarse grid search to identify the best parameter choices for each instrument. We have updated manuscript on lines 781-785 to reflect this explanation, reproduced below:

“We selected default parameters separately for Bionano Irys and Bionano Saphyr instruments based on the tendency for the newer Saphyr instrument to have less directional uncertainty and a lower rate of label collapse. We selected default values for each instrument through a coarse grid search strategy and manual examination of data with known alignments.”

The default numbers of labels for alignment, which are addressed in text and in Supplemental Table 5 were based primarily on the difference in labeling density between BspQI and DLE1. DLE1 has a reference label density 1.7x larger than BspQI. As a result, contigs generated from DLE1 labeling have a higher density of labels per base. We based the minimum length for alignment results based on this, and again used cases where breakpoint graph segments had known alignment locations to optimize these parameter choices. We added a brief description: *“The need for different length thresholds is motivated by the different in labeling density between Irys and Saphyr.”* (lines 834-835).

While a comprehensive comparison of the differences between data generated by Irys and Saphyr instruments would be valuable, it is confounded by properties related to the individual sample prep and we consider such an exhaustive analysis to be outside the scope of the paper.

6) “Finally, as a sincere suggestion to improve software usability, please include visualization or succinct textual outputs on the front page of README of the repo to make it friendlier to the new users so they can know what they expect. Please also consider including a docker image that has everything installed and a copy of the test data that contain a full pipeline script end-to-end, including the command of PrepareAA to build the initial breakpoint graph.”

Response: We have made efforts to improve the software usability and documentation. We have added the following to the AmpliconReconstructor README

- 1) Revise README to better explain dependencies and give precise installation instructions. A user can install AR & CycleViz with a small number of pre-provided commands.
- 2) New README section on inputs and outputs for AR – explaining the different file formats and giving an example of the YAML file used as input.
- 3) Included example commands for generating the AA-derived breakpoint graph using PrepareAA
- 4) Example output images included on CycleViz README.

Because the BAM file and the data repo used by AA are large, we felt that we could not create one single Docker image that contained everything needed to generate the GBM39 test data from BAM file all the way to AR reconstruction. Therefore, we had originally provided an example of running AR using a pre-generated graph.

However, we agree that it should be relatively easy to run this test if a user wants to generate the breakpoint graph. As a result, we have dockerized PrepareAA so that it can, in a single container, run both PrepareAA and install and run AmpliconArchitect. In the AR README we have provided an example of a PrepareAA command one can easily modify to generate breakpoint graph, provided they download the BAM file themselves. With that graph generated a user can run the example AR command provided previously, without the pre-generated graph file.

REVIEWERS' COMMENTS:

Reviewer #1 (Remarks to the Author):

The authors have addressed most of my critiques. I think the paper is strong.

Remaining comments / suggested edits:

(1) The coverage data contain many instances of high magnitude copy changes (>5-10 copies) which are not accompanied by a rearrangement. The authors sort of gloss over the "unexplained variance" and attribute it to tumor heterogeneity, but a subclonal variant would not be likely associated with such a high amplitude (i.e. clonal) copy change. (since a low cancer cell fraction would dilute even a high level amplification).

Examples of these unaccounted for high magnitude copy changes include Figure 3, three copy changes at or near GPC5, DGCR8, MAPK1, or Figure S12a.i to the right of NCOA2, the right of HEY1 the left of NDGR1, and the large right most copy number change.

These are most likely missing (clonal, amplified) rearrangements though some might possibly represent amplicon "ends". They are sometimes associated with segments that has a rearrangement entering at the other end ... so presumably a path is entering that segment, and if so, how is it leaving?

Even if these high magnitude copy number changes are subclonal, they are prevalent enough that the associated rearrangements (if they exist) should be well covered by molecules / reads.

Why are they missed by both OM and WGS? In either case, they should have a large impact on the amplicon reconstruction, depending on how these missing data are treated. Merely attributing these to heterogeneity does not give justice to the magnitude of these changes.

Though it would be interesting / satisfying to investigate these missing rearrangements, I agree it is probably outside of the scope of the analysis. Nevertheless the authors should highlight these shortcomings as they are important both methodologically / analytically and from the standpoint of technology, and obviously are important directions for future work.

(2) The term "copy number ratio" appears sporadically in the text and rebuttal, but it isn't defined. It appears central to the method ...

"Given the graph of linked scaffolds, AR searches for paths in the graph which conform to the copy number ratios in the breakpoint graph." (line 926)

Presumably these are tumor / normal ratios? The tumor normal ratio is a bit of an anachronism from microarray / CGH days a la "CBS".

The authors should describe how these ratios are converted to integer copy numbers. eg using

ploidy and purity? AA seems to deal in integer multiplicities, so it is important to know how these are determined from the "analog" coverage data from OM or NGS. This conversion is also important to determine how precisely the reconstructions explain the coverage data (related to issue (1)), which is still a bit opaque.

Reviewer #2 (Remarks to the Author):

The authors have comprehensively addressed my comments and questions, and the manuscript is in great shape for publication. In addition to the work here, I look forward to seeing AR applied to more (and primary patient) data in the future.

Reviewer #3 (Remarks to the Author):

I would like to thank the authors for taking the efforts to revise the paper. The various issues that I raised earlier are now clarified, except for one place - line 815, as I mentioned before, why is there a minus sign after $P(a)$, i.e. $P(a) - \dots$, shouldn't it be an equal sign instead? Other than that, I support the publication of this work.

Haibao Tang

Reviewer #1 (Remarks to Author):

The authors have addressed most of my critiques. I think the paper is strong.

We thank the reviewer for their thoughtful critiques and valuable suggestions regarding this work.

1. The coverage data contain many instances of high magnitude copy changes (>5-10 copies) which are not accompanied by a rearrangement. The authors sort of gloss over the "unexplained variance" and attribute it to tumor heterogeneity, but a subclonal variant would not be likely associated with such a high amplitude (i.e. clonal) copy change. (since a low cancer cell fraction would dilute even a high level amplification).

Examples of these unaccounted for high magnitude copy changes include Figure 3, three copy changes at or near GPC5, DGCR8, MAPK1, or Figure S12a.i to the right of NCOA2, the right of HEY1 the left of NDGR1, and the large right most copy number change.

These are most likely missing (clonal, amplified) rearrangements though some might possibly represent amplicon "ends". They are sometimes associated with segments that has a rearrangement entering at the other end ... so presumably a path is entering that segment, and if so, how is it leaving?

Even if these high magnitude copy number changes are subclonal, they are prevalent enough that the associated rearrangements (if they exist) should be well covered by molecules / reads.

Why are they missed by both OM and WGS? In either case, they should have a large impact on the amplicon reconstruction, depending on how these missing data are treated. Merely attributing these to heterogeneity does not give justice to the magnitude of these changes.

Though it would be interesting / satisfying to investigate these missing rearrangements, I agree it is probably outside of the scope of the analysis. Nevertheless the authors should highlight these shortcomings as they are important both methodologically / analytically and from the standpoint of technology, and obviously are important directions for future work.

We appreciate this comment as it is a very good explanation of the issue, and we agree that it would be inaccurate to only describe the possibility that these are subclonal events.

Most of the regions the reviewer is referring to, near GPC5, DGCR8, and MAPK1, the right of HEY1, and the left of NDGR1 are areas of lower copy number than the amplicon itself. These particular regions are likely not borne on the amplicon and thus we do not expect to observe them. They may still have an elevated copy number due to abnormally ploidy, yet not be carried on the amplicon. We believe that if an amplicon is joined to these regions (enters) then it is integrated there (Supplemental Figure 15) and that it does not leave again.

There are still some regions of additional CN gain in K562 which AR does not explain compared to the baseline level of the amplicon. As a response to this, in the discussion we have added a very explicit statement, specifically highlighting the example provided by the reviewer in K562 to illustrate how this occurrence reflects a limitation of our current methods.

“Additional copy number changes in K562 near BCR and ABL1 which are not directly explained by the amplicon through edges identified by NGS reveal limitations to our current method or possible inaccuracies. Such cases may indicate additional amplicon segments outside the regions reconstructed by AR, suggesting that the true amplicon structure may extend beyond the regions we have captured.” (line 528)

The argument we make about heterogeneity is slightly different than the standard issue with subclonal populations in a tumor, particularly since these are cell lines. It is not necessary that heterogeneous events occur in a subclonal fraction of the population. For instance, there may be two forms of this amplicon present, both with the same overall segment copy numbers but in different orders. For instance, one copy may take a path through the unexplained segments in one way, and another form may take a path through the unexplained segments in a different way. This will create heterogeneity in both the WGS and OM data which is difficult to resolve with current methods, but maintain an elevated copy number consistent with the amplicon copy number as a whole. Throughout the manuscript we have modified the text to more clearly state these related concepts of “structural” or “amplicon” heterogeneity, as opposed to tumor heterogeneity.

While these cases are concerning, we would expect such problematic regions to create unexplained gaps in the OM data between regions corresponding to rearrangements which are non-variable and consistent between OM and NGS. We do not observe these cases in K562, and thus we believe that we are capturing a core conserved dominant focal amplification, which may indeed have some missing variably-ordered regions flanking it.

We do agree that completely resolving these cases are outside the scope of the current work as we do not have a robust way to identify and measure them. But reducing these kinds of cases is clearly a target for future analysis.

2. The term "copy number ratio" appears sporadically in the text and rebuttal, but it isn't defined. It appears central to the method ...

"Given the graph of linked scaffolds, AR searches for paths in the graph which conform to the copy number ratios in the breakpoint graph." (line 926)

Presumably these are tumor / normal ratios? The tumor normal ratio is a bit of an anachronism from microarray / CGH days a la "CBS".

The authors should describe how these ratios are converted to integer copy numbers. eg using ploidy and purity? AA seems to deal in integer multiplicities, so it is important to know how these are determined from the "analog" coverage data from OM or NGS. This conversion is also important to determine how precisely the reconstructions explain the coverage data (related to issue (1)), which is still a bit opaque.

The reviewer raises an important point about the terminology we use to describe our examination of ratios of copy numbers between amplified segments in a breakpoint graph.

These are not in fact tumor / normal ratios – a clarification we believe we have now resolved with the changes listed below. Instead these are the ratios of copy numbers between amplified segments in the breakpoint graph.

Within reason, this allows us to bypass estimation of purity – as the ratios of graph segment copy numbers are independent of dilution or scaling (i.e. $x : y = a*x : a*y$). The ratios are computed with respect to the amplified segment with the lowest elevated copy number (Methods, “Finding reconstructions in the linked scaffold graph”). This ratio then allows us to specify a capacity for the number of times a segment may appear in the amplicon, $\frac{x}{y}$, as compared to the number of times the segment with the lowest copy number appears.

AA does not deal in integer copy numbers when reporting the estimated graph segment CN. Indeed, AR uses integer multiplicities for the number of times a single segment may appear in an amplicon, however the functions which constrain the maximum integer count need not necessarily set an integer as the upper limit. This is present in the text in equation 9. Thus the conversion of “analog” coverage to digital is done by way of setting a maximum capacity for the number of times a segment may appear, but that capacity is not necessarily an integer.

To help clarify the terminology, we have made some changes to the manuscript:

We have cleaned up the language around “ratios” on lines 270 and 291 to specifically refer to the ratios between amplified segment copy numbers in the graph.

We have cleaned up the issue on line 892 (formerly 926) the reviewer pointed out to more clearly read

“Given the graph of linked scaffolds, AR searches for paths in the graph which conform to the ratio of estimated copy numbers between the graph’s amplified segments.”

Reviewer #2 (Remarks to the Author):

The authors have comprehensively addressed my comments and questions, and the manuscript is in great shape for publication. In addition to the work here, I look forward to seeing AR applied to more (and primary patient) data in the future.

We thank the reviewer for this comment, and for their helpful critiques during this process.

Reviewer #3 (Remarks to the Author):

I would like to thank the authors for taking the efforts to revise the paper. The various issues that I raised earlier are now clarified, except for one place - line 815, as I mentioned before, why is there a minus sign after $P(a)$, i.e. $P(a) - \dots$, shouldn't it be an equal sign instead? Other than that, I support the publication of this work.

Haibao Tang

We again would like to thank the reviewer for his exceptionally close reading of the manuscript.

The reviewer is correct that there was a mistake in how the equation appeared. We have corrected this error in the final version of the manuscript.